# Inference and Sampling for Archimax Copulas

**Yuting Ng**[*]
Duke University
yuting.ng@duke.edu

**Ali Hasan**[*]
Duke University
ali.hasan@duke.edu

**Vahid Tarokh**
Duke University
vahid.tarokh@duke.edu

## Abstract

Understanding multivariate dependencies in both the bulk and the tails of a distribution is an important problem for many applications, such as ensuring algorithms are robust to observations that are infrequent but have devastating effects. Archimax copulas are a family of distributions endowed with a precise representation that allows simultaneous modeling of the bulk and the tails of a distribution. Rather than separating the two as is typically done in practice, incorporating additional information from the bulk may improve inference of the tails, where observations are limited. Building on the stochastic representation of Archimax copulas, we develop a non-parametric inference method and sampling algorithm. Our proposed methods, to the best of our knowledge, are the first that allow for highly flexible and scalable inference and sampling algorithms, enabling the increased use of Archimax copulas in practical settings. We experimentally compare to state-of-the-art density modeling techniques, and the results suggest that the proposed method effectively extrapolates to the tails while scaling to higher dimensional data. Our findings suggest that the proposed algorithms can be used in a variety of applications where understanding the interplay between the bulk and the tails of a distribution is necessary, such as healthcare and safety.

## 1 Introduction

Modeling dependencies between random variables is a central task in statistics, and understanding the dependence between covariates throughout the distribution is important in characterizing a distribution outside of its areas of highest density. For example, in machine learning contexts, a major topic of interest lies in enforcing dependencies between covariates, such as spatial dependence in convolutional neural networks or temporal dependence in recurrent neural networks. Copulas are functions that model the joint dependence of random variables, and they have been successfully employed in a variety of practical modeling settings due to their ease of use and intuitive properties.

Moreover, since copulas are used to represent cumulative distribution functions (CDFs), they have been particularly useful in situations where *tail events* are important – events that have high impact but low probability. For example, in computer vision applications, local dependence modeled by convolutions may be sufficient for the bulk of the data, but for data in the tails of the distribution non-local dependencies may be present and need to be modeled. Current successful applications of copulas have largely been limited to a few basic parametric families and low dimensional settings,

---

[*]Contributed equally. This work was supported in part by the Air Force Office of Scientific Research under award number FA9550-20-1-0397.

preventing their widespread adoption in settings where the dependencies are complicated or the dimension is large.

Archimax copulas are a class of copulas that merges a tractable form with sufficient expressiveness. Archimax copulas effectively balance the representation of data within the bulk of the distribution while extrapolating to the tails by combining the tractability of *Archimedean* copulas with the tail properties of *extreme-value copulas*.

Notably, they remove the simplified symmetry assumption among covariates that is present in Archimedean copulas and the max-stable property of extreme-value copulas. The use of Archimax copulas has resulted in better fit to data in applications such as healthcare [71] and hydrology [3, 16].

However, existing computational methods do not allow feasible inference and sampling for Archimax copulas, preventing their widespread use. Therein lies the motivation behind this work. We construct efficient and flexible inference and sampling methods using deep learning techniques and discuss how they compare to traditional means of density estimation that use existing copula methods and deep generative models. In addition, we provide numerical studies where the proposed Archimax techniques extrapolate to the tails better than existing methods.

## 1.1 Related work

Modeling distributions is a major task in machine learning, with techniques such as generative adversarial networks (GANs) [19], normalizing flows (NFs) [78], and variational autoencoders (VAEs) [54] being the major developments for representing complex distributions.

However, these techniques are largely used for modeling the bulk of a distribution and may not extrapolate well to out-of-distribution samples or samples within the tails, as discussed in [60]. Some methods have been proposed to represent only the tails, for example in [1, 7]. However, they disregard the information in the bulk and only focus on the tails. Recently, Bhatia et al. [6] considered combining GANs with extreme-value theory (EVT). However, this and the above methods are used only for sampling, and do not provide a way of quantifying the dependence of the observation.

Copulas are an important technique for representing distribution functions since they allow easy separation of the marginals and the joint dependence structure of a distribution. Copulas have been applied in machine learning wherein techniques from machine learning have been used in conjunction with traditional copula theory to model more general classes of densities with examples of such work found in [63, 75, 88], please see Appendix A.6 and A.7 for more background on copulas and application of copulas in machine learning. However, these have generally focused on simplified assumptions such as a symmetric dependence between covariates, or a hierarchy of bivariate dependencies. Moreover, these do not extrapolate to tail distributions and do not readily appear to generalize to high dimensions.

With regards to Archimax copulas, several theoretical works have been proposed analyzing the distributions [11, 14, 73]. In Chatelain et al. [16], the authors proposed a method for inferring a stable tail dependence function (stdf) when given an Archimedean generator. However the method assumes knowledge of the Archimedean generator or infers a one-parameter Archimedean generator from pairwise Kendall's taus. Past applications of Archimax copulas were of low dimensions, such as dimensions 2, 3 and 3 in the studies of river flow rates [3], rainfall [16] and nutrient intake [71].

**Our contributions**  We propose methods for filling the gaps in the existing literature by:

1. Developing methods for inferring both the Archimedean generator and stdf;

2. Developing methods for sampling from Archimax copulas;

3. Providing flexible representations for both the radial and spectral components.

Specifically, deep generative models are used to represent the distributions of the radial and spectral components. By taking an expectation, these characterize the Archimedean generator and stdf. The code for this paper is available at `https://github.com/yutingng/gen-AX`.

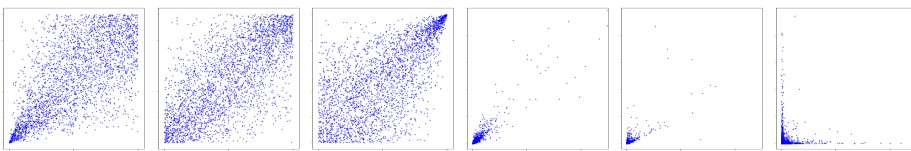

Figure 1: Examples of different radial envelopes and asymptotic dependencies.

## 2 Background

Copulas are given by separating the marginal distributions from the joint dependence of a random variable. Specifically, consider

$$F(\mathbf{x}) = C(F_1(x_1), \cdots, F_d(x_d)), \tag{1}$$

where $F$ is a $d$-variate cumulative distribution function (CDF) of the random variable $\mathbf{X} = (X_1, \cdots, X_d) \in \mathbb{R}^d$, $F_j$ is the $j$th univariate margin, and $C$ is the copula describing the dependence between the uniform random variables $\mathbf{U} = (U_1, \cdots, U_d) = (F_1(X_1), \cdots, F_d(X_d)) \in [0,1]^d$. Moreover, if the marginals $F_j$ are continuous, then the copula $C$ is unique [86].[2]

As discussed in the introduction, Archimax copulas describe a generalization of Archimedean and extreme-value copulas. They are defined as:

**Definition 1 (Archimax copula)** *An Archimax copula is given by*

$$C(\mathbf{u}) = \varphi(\ell(\varphi^{-1}(u_1), \cdots, \varphi^{-1}(u_d))), \tag{2}$$

*where $\varphi : [0, \infty) \to [0, 1]$ is an* Archimedean generator *and $\ell : [0, \infty)^d \to [0, \infty)$ is a* stable tail dependence function (stdf) *[10, 14, 73].*

From the definition, the Archimax copula is completely characterized by these two functions and the objective during inference lies in estimating both the *stdf* and the *Archimedean generator*. Both functions have specific properties that must be fulfilled in order to obtain a valid Archimax copula.

The *stdf* is defined as:

**Definition 2 (stable tail dependence function (stdf))** *A $d$-variate stdf $\ell : [0, \infty)^d \to [0, \infty)$ is given by:*

$$\ell(\mathbf{x}) = d \int_{\Delta_{d-1}} \max_{j \in \{1, \cdots, d\}} \{x_j w_j\} \, \mathrm{d}F_{\mathbf{w}}(\mathbf{w}) = d \, \mathbb{E}_{\mathbf{W}} \left[ \max_{j \in \{1, \cdots, d\}} \{x_j W_j\} \right], \tag{3}$$

*with a spectral random variable $\mathbf{W} \in \Delta_{d-1}$ satisfying moment constraints $\mathbb{E}_{\mathbf{W}}[W_j] = 1/d$ for $j \in \{1, \cdots, d\}$ [22, 47, 82].*

Intuitively, the *stdf* dictates the asymptotic dependence between covariates, with examples given in Figure 1. Notably, the definition completely relies on the distribution of $\mathbf{W}$, and the *stdf* is homogeneous of order one, i.e. $\ell(cx_1, \cdots, cx_d) = c \, \ell(x_1, \cdots, x_d)$ for $c > 0$.

On the other hand, the *Archimedean generator* is defined by:

**Definition 3 (Archimedean generator)** *An Archimedean generator $\varphi : [0, \infty) \to [0, 1]$ is the Williamson $d$-transform of a random variable $R > 0$:*

$$\varphi(x) = \mathcal{W}_R(x) = \int_x^\infty \left(1 - \frac{x}{r}\right)^{d-1} dF_R(r) = \mathbb{E}_R \left[ \left(1 - \frac{x}{R}\right)_+^{d-1} \right], \tag{4}$$

*where $(y)_+ := \max(0, y)$ for $y \in \mathbb{R}$ [72, 97].*

The Archimedean generator has a one-to-one correspondence with the distribution of $R$ [72] and dictates the shape of the radial envelope applied across all covariates, with examples given in Figure 1.

---

[2]notation: random variables in uppercase, observations in lowercase, scalars are not bold, vectors are bold.

With these definitions in mind, we leverage Charpentier et al. [14, Theorem 3.3] for inference and sampling, which established that any random vector

$$\mathbf{X} \stackrel{d}{=} R \times (S_1, \cdots, S_d), \ \mathbf{X} \in [0, \infty)^d, \tag{5}$$

has dependence that follows an Archimax copula, where $R$ and $\mathbf{S}$ are independent and known as the *radial* and *simplex* components, with supports $R > 0$, $\mathbf{S} \in [0, 1]^d$, $\ell(\mathbf{S}) \leq 1$. The marginals of $\mathbf{X}$ is given by $\varphi$ such that the random vector

$$\mathbf{U} \stackrel{d}{=} (\varphi(X_1), \cdots, \varphi(X_d)), \ \mathbf{U} \in [0, 1]^d, \tag{6}$$

follows an Archimax copula. Moreover, every Archimax copula given in (2) has a decomposition given by (5) and (6). The *radial* component $R$ is the same $R$ in Definition 3, and the *simplex* component $\mathbf{S}$ has a one-to-one correspondence with the *spectral* component $\mathbf{W}$ from Definition 2. Therefore, representing the $R$ and $\mathbf{W}$ in Definitions 3 and 2 respectively provides complete understanding of the distribution.

## 3 Method

As established in Section 2, the *stdf* and *Archimedean generator* are expectations of functions of the *spectral* component $\mathbf{W}$ and *radial* component $R$ respectively. We first model these expectations where $\mathbf{W}$ and $R$ are discrete random variables with finite support [31, 34]. We then let them be outputs of generative networks in the limit of infinite support. We begin by describing the inference algorithm for the stdf followed by the inference algorithm for the Archimedean generator. Through sampling the stdf we define the relationship between $\mathbf{S}$ and $\mathbf{W}$ which we make use of in the inference for the Archimedean generator. We finally show how the representations we use can be easily adapted for sampling from Archimax copulas. A flow chart describing the relationship between stdf and Archimedean generator parameter estimation is in Figure 2.

> **Update $\ell_\theta$: Asymptotic Dependence**
> Compute $\xi(\mathbf{u}, \mathbf{x}) = \min_{j \in \{1, \ldots, d\}} \varphi_\theta^{-1}(u_j)/x_k$
> Maximize $\log -\varphi_\theta'(\xi(\mathbf{u}, \mathbf{x})\ell_\theta(\mathbf{x})) + \log \ell_\theta(\mathbf{x})$

> **Update $\varphi_\theta$: Radial Envelope**
> Sample $\ell_\theta(\mathcal{S})$, compute $\mathcal{R}, \mathcal{T}$
> Minimize $\sum_{i \in \{1, \ldots, |\mathcal{T}|\}} (\varphi_\theta(t_i) - w_i)^2$

Figure 2: Flow chart describing relationship between stdf and Archimedean generator estimation.

### 3.1 Stable tail dependence function inference and sampling

For this section, we suppose that the Archimedean generator $\varphi(\cdot)$ is known and the goal is to infer or sample from the stdf.

#### 3.1.1 Inference for stable tail dependence function

We consider a stdf following Definition 2, where $\ell$ is specified by the spectral decomposition with spectral component $\mathbf{W}$ [22, 47, 82]. To summarize the inference for the stdf: we first establish a representation of the stdf through the empirical expectation of samples $\mathbf{w}$ from a generative network $G_\mathbf{W}$. We then define the likelihood of transformed data observations and optimize the parameters of $G_\mathbf{W}$ to maximize this likelihood.

Following Chatelain et al. [16], when given the Archimedean generator $\varphi$ we can define the transformation $\xi(\mathbf{u}, \mathbf{x})$ for an observation $\mathbf{u}$ and a pseudo-observation $\mathbf{x} = (x_1, \cdots, x_d) \in \Delta_{d-1}$:

$$\xi(\mathbf{u}, \mathbf{x}) = \min\{\varphi^{-1}(u_1)/x_1, \cdots, \varphi^{-1}(u_d)/x_d\}, \tag{7}$$

a transformation used for estimating extreme-value copulas [10, 79].

In the case of Archimax copulas, and by using the homogeneity property of the stdf (3), the CDF of the random variable $\xi(\mathbf{U}, \mathbf{x})$ is expressed as:

$$P(\xi(\mathbf{U}, \mathbf{x}) \leq x) = 1 - P(\xi(\mathbf{U}, \mathbf{x}) > x) = 1 - C(\varphi(x\mathbf{x})) = 1 - \varphi(x\ell(\mathbf{x})). \tag{8}$$

The full derivation is given in Appendix A.2.1.

Following Hasan et al. [38], we differentiate the CDF of $\xi(\mathbf{U}, \mathbf{x})$ with respect to $x$ and obtain the log-likelihood of the transformed observation as:

$$\log \mathcal{L}(x; \varphi, \ell) = \log\left(-\varphi'(x\,\ell(\mathbf{x}))\right) + \log(\ell(\mathbf{x})). \tag{9}$$

Recalling Definition 2, let $\ell_\theta$ be parameterized by a generative network $G_{\mathbf{W}}$ such that:

$$\ell_\theta(\mathbf{x}) = \frac{d}{l} \sum_{k=1}^{l} \left[ \max_{j \in \{1, \cdots, d\}} x_j w_{kj} \right], \tag{10}$$

where $\mathbf{w}_k = (w_{k1}, \cdots, w_{kd})$ for $k = \{1, \cdots, l\}$ are $l$ samples from $G_{\mathbf{W}}$ with dimension $d$ and output activation given by the Softmax$(\cdot)$ function to respect the support $\mathbf{W} \in \Delta_{d-1}$. The moment constraints $\mathbb{E}_{\mathbf{W}}[W_j] = 1/d$ for $j = \{1, \cdots, d\}$ are scale conveniences and not necessities [31]. To approximate the moment constraints, we penalize the residual $\sum_{j=1}^{d} (\sum_{k=1}^{l} w_{kj}/l - 1/d)^2$.

We may then train the generative network $G_{\mathbf{W}}$ such that $\ell_\theta$ approximates $\ell$ by minimizing the negative log-likelihood of transformed observations $\xi_(\mathbf{u}, \mathbf{x})$, with $\mathbf{x}$ uniformly sampled on the unit simplex $\Delta_{d-1}$. The full technique is presented in Algorithm 1 in Appendix A.1.

The inverse $\varphi^{-1}$ in the computation of $\xi(\mathbf{u}, \mathbf{x})$ in (7) can be computed numerically by Newton-Raphson. The derivative $\varphi'$ in the computation of $\log \mathcal{L}(x; \varphi, \ell)$ in (9) can be calculated explicitly from Definition 3 as:

$$\varphi'(x) = \mathbb{E}_R \left[ (d-1) \left( -\frac{1}{R} \right) \left( 1 - \frac{x}{R} \right)_+^{d-2} \right]. \tag{11}$$

Putting these computations together results in the likelihood of the stdf given the generator.

### 3.1.2 Sampling the simplex component

We now consider sampling the simplex component $\mathbf{S}$. We noted in the introduction that there is a one-to-one relationship between $\mathbf{S}$ and $\mathbf{W}$. Using this relationship and the result of Charpentier et al. [14], the stdf may also be written in terms of the simplex component through the following definition:

**Definition 4 (survival distribution of simplex component)** *The survival distribution function (SDF) of the simplex component $\mathbf{S}$ is:*

$$P(\mathbf{S} > \mathbf{s}) = \max(0, 1 - \ell(s_1, \cdots, s_d))^{d-1}, \tag{12}$$

which relates to the so-called generalized Pareto copulas defined as:

**Definition 5 (generalized Pareto copula)** *A generalized Pareto copula is defined as the copula of a generalized Pareto distribution with support on $(-\infty, 0]^d$ and CDF specified by $\ell$ as:*

$$P(X_1 < x_1, \cdots, X_d < x_d) = \max(0, 1 - \ell(-x_1, \cdots, -x_d)). \tag{13}$$

*Moreover, it has stochastic representation as:*

$$\mathbf{X} \stackrel{d}{=} -\left( \frac{U}{W_1}, \cdots, \frac{U}{W_d} \right), \tag{14}$$

*where $U$ is uniformly distributed on $[0, 1]$, independent of spectral component $\mathbf{W} = (W_1, \cdots, W_d)$ from the spectral decomposition of $\ell$ in Definition 2 [9, 29].*

Therefore, we can frame the sampling of the simplex component through the sampling from a generalized Pareto copula, such that, given samples $(x_{11}, \cdots, x_{1d}), \cdots, (x_{(d-1)1}, \cdots, x_{(d-1)d})$ of size $d-1$ from a generalized Pareto distribution with CDF in (13), we may obtain a sample of $\mathbf{s}$, where coordinates $s_j$ for $j = \{1, \cdots, d\}$ are computed as [14]:

$$s_j = -\max(x_{1j}, \cdots, x_{(d-1)j}). \tag{15}$$

The coordinate-wise maxima is taken over $d-1$ samples to get the $d-1$ exponent in the SDF of $\mathbf{S}$, corresponding to the $d-1$ exponent in the expression of the Archimedean generator $\varphi$.

Due to our convenient representation of $\ell$ as an expectation of a function of $\mathbf{W}$ from generative network $G_{\mathbf{W}}$, we are able to sample a generalized Pareto distribution from its stochastic representation. The full technique is presented in Algorithm 2 in Appendix A.1.

## 3.2 Archimedean generator inference and sampling

We now assume that the stdf is known and the goal is to infer the Archimedean generator. We consider a general representation of the Archimedean generator following Definition 3, where $\varphi$ is a $d-$monotone function specified by the Williamson $d-$transform of the radial component $R$ [14, 72].

To provide an outline for the overall approach: we first consider the so-called Kendall distribution, which describes an integral transform of the copula. We then use the fact that the empirical Kendall distribution converges to the true Kendall distribution as the empirical copula converges to the true copula, providing a means of estimation. The Kendall distribution is formally defined as:

**Definition 6 (Kendall distribution)** *Let $C$ be a copula, and let the CDF of the random variable* $\mathbf{U} \in [0,1]^d$ *be the copula $C$. Define the random variable $W := C(\mathbf{U})$. The Kendall distribution of $C$ is the multivariate probability integral transform given by the CDF of $W$:*

$$K(w) = P(C(\mathbf{U}) \leq w), \quad w \in [0,1]. \tag{16}$$

In the case of Archimax copulas, using the stochastic representation (5) and the homogeneity property of the stdf (3), the Kendall distribution is expressed as:

$$K(w) = P(\varphi(R\,\ell(S_1, \cdots, S_d)) \leq w). \tag{17}$$

The full derivation is provided in Appendix A.3. The empirical Kendall distribution $K_n$ for $n$ given observations $(u_{11}, \cdots, u_{1d}), \cdots, (u_{n1}, \cdots, u_{nd})$ is defined as:

$$K_n(w) = \frac{1}{n} \sum_{i=1}^{n} \mathbb{1}\{w_i \leq w\}, \tag{18}$$

where

$$w_i = \frac{1}{n+1} \sum_{k=1}^{n} \mathbb{1}\{u_{k1} < u_{i1}, \cdots, u_{kd} < u_{id}\}. \tag{19}$$

In the case of symmetric dependence in the non-parametric inference of Archimedean copulas [34], $\mathbf{S}$ is uniformly distributed on the simplex $\Delta_{d-1}$ and $\ell(S_1, \cdots, S_d) = S_1 + \cdots + S_d \equiv 1$. Then, $R$ is discrete with the cardinality of the support the same as $W$ and $K_n$ is the Kendall distribution of a unique Archimedean copula [34].

In the non-parametric inference of Archimax copulas, we define the random variables:

$$Z := \ell(S_1, \cdots, S_d) \quad \text{and} \quad T := RZ \tag{20}$$

where $T$ is a discrete random variable with support the same size as $W$ and $K_n$ is the Kendall distribution of an Archimax copula. Note that $R$ and $Z$ are independent as $R$ and $\mathbf{S}$ are independent.

Using $K_n$, we reconstruct the support of $R$, given $\ell$ and the support of $\mathbf{S}$, which in turn provides the support of $Z$. With the final objective of letting $R$ and $\mathbf{S}$ be independent and identically distributed (iid) outputs of generative networks, we first (linearly) interpolate $K_n$ to be equispaced, such that each $w_i$ has the same probability $k_i = 1/(n_r n_z)$, where $n_r$ and $n_z$ are the chosen sizes of supports for $R$ and $Z$. We describe the reconstruction procedure for the general case with non-iid random variables including cases where $r_j z_l = r_{j'} z_{l'}$ in Appendix A.3. The sizes of supports $n_r$ and $n_z$ are chosen empirically with examples given in Appendix B.1.

Suppose the supports of the distributions of $R, Z, W, T$ are finite and respectively denoted by: $\mathcal{W} = \{w_1, w_2, \cdots, w_{n_r n_z}\}$, $\mathcal{R} = \{r_1, r_2, \cdots, r_{n_r}\}$, $\mathcal{Z} = \{z_1, z_2, \cdots, z_{n_z}\}$, $\mathcal{T} = \{r_j z_l : r_j \in \mathcal{R}, z_l \in \mathcal{Z}\} = \{t_1, t_2, \cdots, t_{n_r n_z}\}$, where the $n_r n_z$ elements of $\mathcal{W}$ are sorted in decreasing (non-increasing) order, and the $n_r n_z$ elements of $\mathcal{T}$ are sorted in increasing (non-decreasing) order. This reverse ordering is due to $\varphi$ being a decreasing function.

We minimize the mean sum of square residuals motivated by the uniform convergence of the empirical process $\sqrt{n}(K_n - K)$ as $n \to \infty$ established by Barbe et al. [4]:

$$\frac{1}{n_r n_z} \sum_{i=1}^{n_r n_z} (w_i - \varphi_\theta(t_i))^2 \tag{21}$$

where, following Definition 3,

$$\varphi_\theta(t_i) = \mathcal{W}_\mathcal{R}(t_i) = \frac{1}{n_r} \sum_{j=1}^{n_r} \left( 1 - \frac{t_i}{r_j} \right)_+^{d-1}. \tag{22}$$

The finite support assumption of $R$ and $Z$ is not necessary, and we consider a modification where the supports $\mathcal{R}, \mathcal{Z}$ are specified by samples from generative networks. The main objective is to learn the parameters of generative network $G_R$ given the empirical Kendall distribution $K_n$ and samples of $Z$. Since scaling the support $\mathcal{R}$ by a constant $c > 0$ does not change the copula, we add a regularization term for $\mathbb{E}_R[R] = 1$. The algorithm can be understood as an alternating minimization algorithm, where the map between $\mathcal{R}$ and $\mathcal{Z}$ to $\mathcal{W}$ via $\mathcal{T}$, and the support $\mathcal{R}$ are updated in an alternating fashion. The full technique is presented in Algorithm 3 in Appendix A.1.

The learned $G_R$ also provides a source of samples for the radial component $R$, which we use to generate full samples from the Archimax copula.

## 3.3 Inference and sampling for Archimax copulas

We finally summarize the combination of inference and sampling for both the Archimedean and simplex component to obtain the full algorithm for inference and sampling for Archimax copulas. This culminates into an iterative technique that successively updates each component.

### 3.3.1 Inference for Archimax copulas

We initialize with $\varphi(x) = \exp(-x)$ and infer $\ell$ to learn $G_\mathbf{W}$, following Section 3.1. This special combination of $\varphi$ and $\ell$ corresponds to extreme-value dependence with the max-stable property. To aid inference for this initialization step, we pre-process our data to have extreme-value dependence. We do so by computing the *block maxima*, a technique from extreme-value copulas where we group observations and take the coordinate-wise maximas within each group. Specifically, given observations $(x_{11}, \cdots, x_{1d}), \cdots, (x_{n1}, \cdots, x_{nd})$, the block-maximas $(m_{11}, \cdots, m_{1d}), \cdots, (m_{k1}, \cdots, m_{kd})$ for $k$ blocks of size $n/k$ are computed as:

$$m_{ij} = \max(x_{(n/k)(i-1)+1\,j}, \cdots, x_{(n/k)(i)\,j}), \tag{23}$$

for $i = \{1, \cdots, k\}$ and $j = \{1, \cdots, d\}$. To determine the block size, with larger block sizes $n/k$ better approximating extreme-value dependence and more blocks $k$ for more observations, we test the block maximas for extreme-value dependence via the max-stable property using the test by Kojadinovic et al. [55], and select the first $k$ where the null hypothesis of extreme-value dependence is not rejected, starting from $k = n$, with details given in Appendix A.4.1.[3]

Given an estimate of $\ell$ and a learned $G_\mathbf{W}$, we are able to generate many samples of $\mathbf{S}$, thereby providing a way to compute $Z = \ell(\mathbf{S})$. We then infer $\varphi$, learning $G_R$, following Section 3.2. At this point, we repeat the estimation for $G_\mathbf{W}$ with the updated $\varphi$ to improve our estimate for $\ell$. The algorithms may be iterated as needed with suggested convergence criteria such as Cramérvon Mises (CvM) distance between successively estimated copulas.

### 3.3.2 Sampling for Archimax copulas

Given samples of the radial and simplex components, the stochastic representation of Archimax copulas (5) gives a straightforward method for sampling Archimax copulas [14]. Specifically, we sample $\mathbf{S}$ and $R$, then multiply and normalize by $\varphi$; see Algorithm 4 in Appendix A.1 for details.

## 4 Experiments

To understand the modeling and sampling capabilities of the proposed algorithms, we conduct a number of empirical studies. We compare the proposed method to a number of existing copula based and deep learning based methods for density estimation. Our experiments relate to the main focus

---

[3]An alternative initialization scheme based on testing different one-parameter families of Archimedean generators with the log-likelihood of transformed observations $\xi$ is also given in Appendix A.4.1.

of the proposed method where we are interested in understanding the dependencies between the variables in both the bulk and the tail. In that sense, we conduct a number of experiments where we wish to extrapolate to the tail. Further experimental details may be found in Appendix B.

The metric we use to compare the methods is based on the Cramér-von Mises (CvM) statistic, which computes the $L^2$ distance between the empirical copula and the estimated copula. This statistic is commonly used to determine differences between distributions in goodness-of-fit tests [81]. We use a version where we compare the empirical copula of true samples to the empirical copula of generated samples. Specifically, it can be written as: $CvM = \int (C_{*,n}(\mathbf{u}) - C_{\theta,n}(\mathbf{u}))^2 \mathrm{d}\mathbf{u}$, where $C_{*,n}$ is the empirical copula of true samples and $C_{\theta,n}$ is the empirical copula of generated samples [81].

**Inference for Archimedean generator** Our first set of experiments involve inference for the Archimedean generator following the proposed method in Section 3.2. We consider data with dimension $d = 10$ and sample size of $n = 1000$ given the true stdf $\ell$. To the best of our knowledge, our proposed method is the first method for non-parametric inference of flexible Archimedean generators in Archimax copulas. As such, there are no baselines for comparison. Instead, we compute the results in terms of the map $\lambda(w) = \varphi^{-1}(w)/(\varphi^{-1}(w))', w \in (0,1)$ due to its scale invariant property and known asymptotic variance, which was described as a useful metric for how well $\varphi$ is fit in [33, 34]. We evaluated our proposed method on the Clayton (C), Frank (F), Joe (J) and Gumbel (G) generators, representing different radial envelopes, for Kendall's tau of $\tau = \{0.2, 0.5\}$, representing different associations. The stdf comes from the family of negative scaled extremal Dirichlet (NSD), a flexible class of stdf [5]. All estimates were within the asymptotic variance of $\lambda(w), w \in (0,1)$. The next best method is comparing to a Clayton generator estimated by pairwise Kendall's tau [16]. We give the results in terms of MSE to $\lambda$ in Table 1, and plot estimates of $\lambda(w)$ in Figure 5 in Appendix B.1. Additional experiment results, including small sample performance $n = 200$, and choices of support sizes $n_r$ and $n_z$, are in Appendix B.1.

**Inference for stable tail dependence function and sampling for simplex component** We now consider the reverse scenario where we wish to estimate the stdf $\ell$ given the true Archimedean generator $\varphi$. We compute the integrated relative absolute error (IRAE) between the estimated $\ell_\theta$ and the true $\ell$. The IRAE is given by $IRAE(\ell, \ell_\theta) = \frac{1}{|\Delta_{d-1}|} \int_{\Delta_{d-1}} |\ell(\mathbf{x}) - \ell_\theta(\mathbf{x})|/\ell(\mathbf{x}) \, \mathrm{d}\mathbf{x}$ [16]. For the same experiment settings as above, results are in Table 1. Additional experiment results, including time taken, are in Appendix B.2.

Table 1: Inference of $\varphi$ given true $\ell$ and inference of $\ell$ given true $\varphi$

|  | C 0.2 | C 0.5 | F 0.2 | F 0.5 | J 0.2 | J 0.5 | G 0.2 | G 0.5 |
|---|---|---|---|---|---|---|---|---|
| MSE $\times 10^{-3}$ [16] | **0.01** | **0.04** | 0.9 | 9 | 1 | 9 | 0.7 | 8 |
| MSE $\times 10^{-3}$ (OURS) | 0.2 | 0.2 | **0.1** | **0.1** | **0.3** | **0.1** | **0.2** | **0.1** |
| IRAE $\pm 0.01$ [16] | 0.05 | 0.11 | 0.04 | 0.04 | 0.05 | 1.00 | 0.06 | 0.15 |
| IRAE $\pm 0.01$ (OURS) | 0.06 | 0.12 | 0.04 | 0.05 | 0.06 | **0.07** | 0.08 | 0.15 |

**Modeling nutrient intake** The USDA studied the nutrient intake of women [92]. One particular task is understanding the dependencies between the intake of different nutrients. We can model this using an Archimax copula to understand the dependencies between the nutrients in the bulk and the tail. To assess this, we fit and compare models from the literature on representing distributions and compute the CvM goodness-of-fit statistic for each model. Specifically, we break our comparison into two different types of models: copula based models and deep network based models. For the copula models, for methods marked with *, we use the $\varphi$ described in Section 3.2 and for methods marked with † we use the $\ell$ described in Section 3.1.2. For the deep generative models, we use standard methods based on the Wasserstein GAN [2], masked autoregressive flow (MAF) [77], and variational autoencoders (VAE) [54]. The results are presented in Table 2. The proposed method (Gen-AX) with the Archimax has the lowest CvM distance among all the competing methods, suggesting that the proposed method is recovering the true dependency structure. The state of the art Clayton Archimax (C-AX) did not perform well possibly due to the difficulty in scaling the single parameter Clayton generator to higher dimension. The Archimedean copula (AC *) possibly benefited from the use of our proposed $\varphi$. We additionally provide examples of samples versus the ground truth in Appendix B.3 for the different methods as well as an explanation of the different abbreviations.

Table 2: Goodness-of-fit to nutrient intake data and 100d NSD copula

| | COPULAS | | | | | | | | DEEP NETS | | | OURS |
|---|---|---|---|---|---|---|---|---|---|---|---|---|
| | GC | RV | CV | DV | AC * | HAC | EV † | C-AX † | GAN | MAF | VAE | Gen-AX *† |
| Nutrient CvM $\times 10^{-3}$ | 0.081 | 0.3 | 0.2 | 0.6 | 0.030 | 0.2 | 0.4 | 0.6 | 0.033 | 0.036 | 0.053 | **0.026** |
| C-NSD CvM $\times 10^{-5}$ | - | - | - | - | - | - | - | - | 16 | **3** | 5 | **3** |

**Extrapolating to extreme rainfall** Archimax copulas were initially developed as a tool to study the behaviour of methods that estimate the joint distribution of extreme events [11]. The extreme-value copula that arises in the limit can be understood from the stdf $\ell$ and the index of regular variation of the Archimedean generator $\varphi$. Unlike extreme-value copulas which emerge from the limiting distribution of extreme events, the motivation for the use of Archimax copulas is to model extreme data, where observations are rare, from a mix of moderately less extreme data, where observations are relatively more abundant.

In this experiment, we consider another realistic dataset which models the monthly rainfall in French Britanny as studied by Chatelain et al. [16]. We are interested in testing how well the proposed method can extrapolate to the extremes from non-extreme data. Specifically, we analyze the monthly rainfall data, which did not pass the test of extreme-value dependence [55, 16].

We first train the models on the full dataset. We then generate many samples from the trained model, compute the block maxima, then estimate the extremal dependence from the block maxima using the CFG estimator [10]. The results are presented in Table 3 where we compare the proposed method to deep generative models. Additional experiment details and results, including plots of samples from the bulk and the extremes, are in Appendix B.4. We did not compare to copula based models since classical copulas are generally Gumbel or independent in the extremes and thus not suitable for this application. However, for the purpose of modeling monthly rainfall without extrapolating to extremes, we compared to a variety of copula based models, including skew-t copulas which is a class of flexible asymmetric copulas [23, 56, 99, 87]. The full details and results are given in Appendix B.4.1.

**Out-of-distribution detection** Using the same realistic dataset as above, we added outliers generated uniformly at random on the unit cube. The AUC and F1 scores for outlier detection based on likelihoods are given in Table 4. Additional experiment details and figures are in Appendix B.5.

Table 3: Goodness-of-fit to dependence in the extremes

| IRAE $\pm 0.01$ | GAN | MAF | VAE | C-AX | OURS |
|---|---|---|---|---|---|
| C-NSD | 0.52 | 0.42 | 0.16 | 0.12 | **0.03** |
| F-NSD | 0.10 | 0.15 | **0.04** | **0.04** | **0.04** |
| J-NSD | **0.03** | 0.48 | **0.03** | 0.08 | **0.03** |
| G-NSD | 0.07 | 0.38 | 0.08 | 0.16 | **0.04** |

Table 4: Out-of-distribution detection

| | MAF | VAE | OURS |
|---|---|---|---|
| AUC | 0.82 | 0.37 | **0.92** |
| F1 | 0.48 | 0.04 | **0.72** |

**High dimensional modeling** We finally consider an experiment where we infer and sample data from a 100-dimensional Clayton-NSD Archimax copula. Scaling to high dimensions is an important property of the proposed method since many existing copula methods fail to scale beyond lower dimensions. As such, we only compare with deep generative models since the existing copula models resulted in numerical errors during optimization. We report the CvM statistic in the second line of Table 2. We additionally provide examples of the samples in Appendix B.6.

## 5 Conclusion

We developed highly flexible and scalable inference and sampling algorithms, facilitating the use of Archimax copulas in practical settings. We experimentally compare to state-of-the-art density modeling techniques, and the results suggest that the proposed method effectively extrapolates to tails while scaling to higher dimensional data. The methods are especially useful in scenarios requiring extrapolations to the tails while also incorporating data from the bulk.

**Limitations and future work** A single Archimedean generator $\varphi$ to describe the radial envelope across all coordinates may not be sufficiently expressive for certain datasets. For these cases, hierarchical Archimax copulas may be more appropriate and a direction of future work [42]. Other directions for future work include modifying the generator architectures to allow modeling of temporal dependence and also application of Archimax copulas to describe dependencies of non-tabular data, such as via graph neural networks [68].

**Potential negative societal impacts** Model misspecification may lead to misspecification of risks, leading to potentially catastrophic outcomes in areas such as healthcare, safety and finance. Risks may be mitigated by confirming a reasonable fit between the observations generated by the model and the data.

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
