# Inference and Sampling for Archimax Copulas: Supplementary Material

## A    Algorithms and Background

### A.1    Algorithms

We provide algorithms to describe the algorithmic contributions in the main paper. Algorithm 1 describes the method for learning the stable tail dependence function (stdf) given the Archimedean generator. Algorithm 2 describes sampling the simplex component with a learned spectral measure. Algorithm 3 describes learning the Archimedean generator with a given stdf. Finally, a sampling algorithm for the full Archimax copula is provided in Algorithm 4 assuming a learned generator and stdf. The code for this paper is available at `https://github.com/yutingng/gen-AX`.

---

**Algorithm 1** Learn stable tail dependence function (stdf)

---

**input** observations $\{\mathbf{u}_i : i = 1, ..., m\}$.
**input** Archimedean generator $\varphi_\theta$.
**initialize** $G_{\mathbf{W}}$.
**do while** loss $= $ NLL $+$ reg **not converged**:
    **sample** $\{\mathbf{x}_j : j = 1, ..., n\}$ from Unif$(\Delta_{d-1})$.
    **compute** $\{\xi(\mathbf{u}_i, \mathbf{x}_j) : i = 1, ..., m, \; j = 1, ..., n\}$ from (7).
    **sample** $\{\mathbf{w}_k : k = 1, ..., l\}$ from $G_{\mathbf{W}}$.
    **compute** $\{\ell_\theta(\mathbf{x}_j) : j = 1, ..., n\}$ from (10).
    **compute** NLL $= \frac{1}{mn} \sum_{i=1,j=1}^{i=m,j=n} \log \mathcal{L}(\xi(\mathbf{u}_i, \mathbf{x}_j); \varphi_\theta, \ell_\theta)$ from (9).
    **compute** reg $= \sum_{j=1}^d (\sum_{k=1}^l w_{kj}/l - 1/d)^2$.
    **descent** argmin$_{G_{\mathbf{W}}}$ NLL $+$ reg.
**end while**
**return** learned $G_{\mathbf{W}}$.

---

**Algorithm 2** Sample simplex component

---

**input** learned $G_{\mathbf{W}}$.
**do while** $\ell_\theta(\mathbf{s}) > 1$:
    **sample** $\{u_i : i = 1, ..., d-1\}$ from Unif$(0, 1)$.
    **sample** $\{(w_{i1}, ..., w_{id}) : i = 1, ..., d-1\}$ from $G_{\mathbf{W}}$.
    **compute** $\{(x_{i1}, ..., x_{id}) : i = 1, ..., d-1\}$ from (14).
    **compute** $\mathbf{s} = (s_1, ..., s_d)$ from (15).
    **compute** $\ell_\theta(\mathbf{s})$ from (10).
**end while**
**return** a sample $\mathbf{s} = (s_1, ..., s_d)$.

---

**Algorithm 3** Learn Archimedean generator

---

**input** Kendall observations $\mathcal{W} = \{w_i : i = 1, ..., n_r n_z\}$.
**input** learned $G_Z$ such that $Z := \ell_\theta(\mathbf{S})$, $\mathbf{S}$ sampled from learned $G_{\mathbf{W}}$ with Algorithm 2.
**initialize** $G_R$.
**sort** $\mathcal{W}$ in decreasing (non-increasing) order.
**do while** MSE $= \sum_{i=1}^{n_r n_z} (w_i - \varphi_\theta(t_i))^2/(n_r n_z) > \epsilon$ :
    **sample** $\mathcal{R} = \{r_1, ..., r_{n_r}\}$ from $G_R$.
    **sample** $\mathcal{Z} = \{z_1, ..., z_{n_z}\}$ from $G_Z$.
    **compute** $\mathcal{T} = \{r_j z_l, \; r_j \in \mathcal{R}, z_l \in \mathcal{Z}\}$.
    **sort** $\mathcal{T}$ in increasing (non-decreasing) order.
    **compute** $(\varphi_\theta(t_1), ..., \varphi_\theta(t_{n_r n_z}))$ from (22).
    **descent** argmin$_{G_R} \sum_{i=1}^{n_r n_z} (w_i - \varphi_\theta(t_i))^2/(n_r n_z) + (\sum_{j=1}^{n_r} r_j/n_r - 1)^2$.
**end while**
**return** learned $G_R$.

---

---

**Algorithm 4** Sample Archimax copulas

---

    **input** learned $G_R$, $G_{\mathbf{W}}$
    **sample** $r, r_1, \cdots, r_{n_r}$ from $G_R$.
    **sample** $\mathbf{s} = (s_1, \cdots, s_d)$ from $G_{\mathbf{W}}$ with Algorithm 2.
    **compute** $\mathbf{u} = (\varphi_\theta(rs_1), \cdots, \varphi_\theta(rs_d))$ from (22).
    **return** a sample $\mathbf{u} = (u_1, \cdots, u_d)$.

---

## A.2    Stable tail dependence function inference and sampling

### A.2.1    Inference for stable tail dependence function

**Pickands transformation**    The Pickands transformation in (7) for a particular test point $\mathbf{x} \in \Delta_{d-1}$ results in the transformed observation $\xi(\mathbf{U}, \mathbf{x})$ with survival distribution function (SDF):

$$
\begin{aligned}
P(\xi(\mathbf{U}, \mathbf{x}) > x) &= P(\min\{\varphi^{-1}(U_1)/x_1, \cdots, \varphi^{-1}(U_d)/x_d\} > x), \\
&= P(\varphi^{-1}(U_1) > xx_1, \cdots, \varphi^{-1}(U_d) > xx_d), \\
&= P(U_1 < \varphi(xx_1), \cdots, U_d < \varphi(xx_1)), && \varphi \text{ is decreasing}, \\
&= \varphi(\ell(xx_1, \cdots, xx_d)), && C(\mathbf{u}) := \varphi(\ell(\varphi^{-1}(\mathbf{u}))), \\
&= \varphi(x\ell(x_1, \cdots, x_d)), && \ell(c\mathbf{x}) = c\ell(\mathbf{x}), c > 0, \\
&= \varphi(x\ell(\mathbf{x})).
\end{aligned}
$$

In the case of extreme-value copulas, $\varphi(x) := \exp(-x)$, $\varphi^{-1}(u) := -\log(u)$ and $C(\mathbf{u}) := \exp(-\ell(-\log(u_1), \cdots, -\log(u_d)))$, such that the $d-$dimensionsal observations distributed according to the extreme-value copula are transformed into $1-$dimensional observations distributed according to an exponential with rate $\ell(\mathbf{x})$. The stdf at a particular test point $\ell(\mathbf{x})$ is then estimated from the mean of the transformed observations, with endpoint corrections, as [79]:

$$
\hat{\ell}(x) = \left( \sum_{i=1}^n -\log\left(\frac{i}{n+1}\right) \right) \Big/ \left( \sum_{i=1}^n \xi(\mathbf{u}_i, \mathbf{x}) \right). \tag{24}
$$

A modification to the Pickands estimator, using the transformation $\log(\xi(\mathbf{U}, \mathbf{x}))$, leads to the CFG estimator [10]:

$$
\log(\hat{l}(\mathbf{x})) = \frac{1}{n} \sum_{i=1}^n \log\left( -\log\left(\frac{i}{n+1}\right) \right) - \frac{1}{n} \sum_{i=1}^n \log(\xi(\mathbf{u}_i, \mathbf{x})). \tag{25}
$$

A modification to the CFG estimator was made by Chatelain et al. [16] for the case of Archimax copulas:

$$
\log(\hat{l}(\mathbf{x})) = \frac{1}{n} \sum_{i=1}^n \log\left( \varphi^{-1}\left(\frac{i}{n+1}\right) \right) - \frac{1}{n} \sum_{i=1}^n \log(\xi(\mathbf{u}_i, \mathbf{x})). \tag{26}
$$

In the main paper, we provide the likelihood of a transformed observation $\xi(\mathbf{u}, \mathbf{x})$ in (9) and directly train the generative network to maximize this likelihood. We do not have an additional endpoint correction step since the stochastic form of $\ell_\theta$ in (10) is a valid stdf. On the other hand, the Pickands and CFG estimators above have corrected endpoints but may not be valid stdfs [38]. In the additional experimental results in Appendix B.2 our estimator, although based on the Pickands estimator, performs better than the Pickands estimator. This may be due to our estimator representing the class of valid stdfs, a phenomenon noted in [30] where projecting to the class of valid stdfs reduced estimation error. This direct method of training may perform better than the alternative method that first estimates $\hat{\ell}$ then trains the generative network to match $\hat{\ell}$. A future direction may be to improve the robustness of our MLE estimator with the CFG modification.

**Architecture of generative network for spectral component**    $G_{\mathbf{W}}$ is a generic multilayer perceptron with layers: (Linear$(d, d_h)$, BatchNorm$(d_h)$, ReLU(), Linear$(d_h, d_h)$, BatchNorm$(d_h)$, ReLU(), Linear$(d_h, d)$, Softmax()), where $d$ is the dimension of the observations and $d_h$ is the

number of nodes in the hidden layers. The number of layers and number of hidden nodes may be modified as needed. Batch normalization, i.e. BatchNorm($\cdot$), greatly helps in preventing samples from being static during training.

**Details of Algorithm 1**    To reduce computational complexity, we use mini-batch gradient descent and a smaller number of samples in the empirical expectations of $\varphi_\theta$ and $\ell_\theta$ during training.

### A.2.2   Sampling the simplex component

**Details of Algorithm 2**    From Definition 4, the marginals of the simplex component are distributed as $\text{Beta}(1, d - 1)$, where $d$ is the dimension. To enforce the marginals, we compute the samples of the empirical copula $\mathbf{u}$, then apply the quantile function, i.e. inverse cumulative distribution function (CDF), to obtain $\mathbf{s}$ such that:

$$s_j = 1 - (1 - u_j)^{\frac{1}{d-1}} \text{ for } j = \{1, \cdots, d\}. \tag{27}$$

Given $n$ observations $(x_{11}, \cdots, x_{1d}), \cdots, (x_{n1}, \cdots, x_{nd})$ the samples of the empirical copula $(u_{11}, \cdots, u_{1d}), \cdots, (u_{n1}, \cdots, u_{nd})$ are coordinate-wise rank-normalized such that:

$$u_{ij} = \frac{1}{n} \sum_{k=1}^{n} \mathbb{1}\{x_{kj} \leq x_{ij}\} \text{ for } i = \{1, \cdots, n\}, j = \{1, \cdots, d\}. \tag{28}$$

### A.3   Archimedean generator inference and sampling

**Kendall distribution function**    The expression of $K(w)$ for Archimax copulas is:

$$
\begin{aligned}
K(w) &= P(C(U_1, \cdots, U_d) \leq w), \\
&= P(\varphi(\ell(\varphi^{-1}(U_1), \cdots, \varphi^{-1}(U_d))) \leq w), && C(\mathbf{u}) := \varphi(\ell(\varphi^{-1}(\mathbf{u})), \\
&= P(\varphi(\ell(RS_1, \cdots, RS_d)) \leq w), && \mathbf{U} \stackrel{d}{=} \varphi(R\mathbf{S}), \\
&= P(\varphi(R\ell(S_1, \cdots, S_d)) \leq w), && \ell(c\,\mathbf{s}) = c\,\ell(\mathbf{s}), c > 0, \\
&= P(\varphi(RZ) \leq w), && Z := \ell(\mathbf{S}), \\
&= P(\varphi(T) \leq w), && T := RZ.
\end{aligned}
$$

**Reconstruction of radial distribution with non-iid random variables and repetition of elements** To provide an outline for the overall approach, given the current estimate of $\mathcal{R}$, we compute the mapping between $\mathcal{R}, \mathcal{Z}$ and $\mathcal{T}$, solve for the probabilities $P_\mathcal{R}$, update the support $\mathcal{R}$, and iterate as needed. The algorithm can be understood as an alternating minimization algorithm, where the map between $\mathcal{R}, \mathcal{Z}$ to $\mathcal{W}$ via $\mathcal{T}$, and the support $\mathcal{R}$ are updated in an alternating fashion.

We initialize by computing $\mathcal{W} = \{w_1, \cdots, w_m\}$ and $P_\mathcal{W} = \{p(w_1), \cdots, p(w_m)\}$ of the empirical Kendall distribution function in (18) and (19), where $m \leq n_r n_z$. We do not perform the additional (linear) interpolation step of the main paper. We then sort $\mathcal{W}$ in decreasing order. We also initialize $\mathcal{R} = \{r_{n_r} = 1, r_j = r_{j+1}\alpha_j \text{ for } j = \{1, ..., n_r - 1\}\}$, where we select $\alpha = (0.9, ..., 0.9)$. The support $\mathcal{Z}$ and probabilities $P_\mathcal{Z}$ are assumed to be given.

Given $\mathcal{Z}$ and the current estimate of $\mathcal{R}$, we compute

$$\mathcal{T} = \{r_j z_l : r_j \in \mathcal{R}, z_l \in \mathcal{Z}\}. \tag{29}$$

We then sort $\mathcal{T}$ in increasing order and compute the ordering

$$(\sigma_r, \sigma_z)(i) : \{1, \cdots, m\} \to \{1, \cdots, n_r\} \times \{1, \cdots, n_z\} \tag{30}$$

defined as a surjective function such that

$$t_i = r_{\sigma_r(i)} z_{\sigma_r(i)}. \tag{31}$$

We solve for the probabilities $P_\mathcal{R} = \{p(r_1), \cdots, p(r_{n_r})\}$ by minimizing the residuals

$$\sum_{i=1}^{m} \left( \left( \sum_{(\sigma_r^{-1}(j), \sigma_z^{-1}(l))=i} p(r_j)p(z_l) \right) - p(w_i) \right)^2. \tag{32}$$

We solve for the support $\mathcal{R} = \{r_1, \cdots, r_{n_r}\}$, by minimizing the residuals

$$\sum_{i=1}^{m}(\varphi_\theta(t_i) - w_i)^2 \tag{33}$$

where, following Definition 3,

$$\varphi_\theta(t_i) = \sum_{j=1}^{n_r} p(r_j)\left(1 - \frac{t_i}{r_j}\right)_+^{d-1}. \tag{34}$$

**Numerical illustration**   We give the details of the algorithm with a simple numerical example.

Consider the supports $\mathcal{R} = (r_1, r_2, r_3) = (1, 2, 3)$, $\mathcal{Z} = (z_1, z_2) = (0.5, 0.75)$ with probabilities $P_\mathcal{R} = (0.4, 0.4, 0.2)$, $P_\mathcal{Z} = (0.25, 0.75)$ and noisy observations $\mathcal{W} = (0.07, 0.19, 0.34, 0.49, 0.67)$, $P_\mathcal{W} = (0.07, 0.33, 0.14, 0.31, 0.15)$.

In this case, $(t_1, t_2, t_3, t_4, t_5) = (0.5, 0.75, 1.0, 1.5, 2.25)$ corresponds to $(r_1 z_1, r_1 z_2, r_2 z_1, r_2 z_2 \cup r_3 z_1, r_3 z_2)$ with probabilities $(0.1, 0.3, 0.1, 0.3 + 0.05, 0.15)$.

For $P_\mathcal{R}$, we solve the overdetermined system of linear equations:

$$\begin{pmatrix} 0.25 & & \\ 0.75 & & \\ & 0.25 & \\ & 0.75 & 0.25 \\ & & 0.75 \end{pmatrix} \begin{pmatrix} p_1 \\ p_2 \\ p_3 \end{pmatrix} = \begin{pmatrix} 0.07 \\ 0.33 \\ 0.14 \\ 0.31 \\ 0.15 \end{pmatrix} \tag{35}$$

and obtain the solution as $(0.43, 0.37, 0.20)$, where the solution has been normalized to sum to 1.

For $\mathcal{R}$, we minimize the residuals in (33). Since scaling such that $\mathcal{R} = \{cr_1, \cdots, cr_{n_r}, c > 0\}$ does not change the copula, we solve for $\mathcal{R}$ in terms of ratios $(\alpha_1, \cdots, \alpha_{n_r-1})$, recursively defined such that $r_{n_r} = 1$ and $r_j = r_{j+1}\alpha_j$ for $j = \{1, \cdots, n_r - 1\}$.

The full technique is presented in Algorithm 5, with code attached in the supplementary material.

---

**Algorithm 5** Estimate radial component

---

   **input** $\mathcal{W} = \{w_i : i = 1, ..., m\}$, $P_\mathcal{W} = \{p(w_i) : i = 1, ..., m\}$.
   **input** $\mathcal{Z} = \{z_l : l = 1, ..., n_z\}$, $P_\mathcal{Z} = \{p(z_l) : l = 1, ..., n_z\}$.
   **initialize** $(\alpha_1, ..., \alpha_{n_r-1})$ for instance $(0.9, ..., 0.9)$.
   **sort** $\mathcal{W}$ in decreasing order.
   **do while** MSE $= \frac{1}{m}\sum_{i=1}^{m}(\varphi_\theta(t_i) - w_i)^2 > \epsilon$ :
      **compute** $\mathcal{R} = \{r_{n_r} = 1, r_j = r_{j+1}\alpha_j$ for $j = 1, ..., n_r - 1\}$.
      **compute** $\mathcal{T} = \{r_j z_l, r_j \in \mathcal{R}, z_l \in \mathcal{Z}\}$.
      **sort** $\mathcal{T}$ in increasing order.
      **compute** $\{(\sigma_r, \sigma_z)(i) : i = 1, ..., m\}$ from (31).
      **solve** $P_\mathcal{R}$ from (32).
      **compute** $(\varphi_\theta(t_1), ..., \varphi_\theta(t_m))$ from (34).
      **solve** $\text{argmin}_\alpha \sum_{i=1}^{m}(w_i - \varphi_\theta(t_i))^2$ such that $\alpha \in (\alpha_l, \alpha_u)$ for instance $(0.01, 1)$.
   **end while**
   **return** Estimated support $\mathcal{R}$ and probabilities $P_\mathcal{R}$.

---

The ratios $\alpha$ may be solved iteratively for a unique solution. In our case, motivated by the uniform convergence of the empirical process $\sqrt{n}(K_n - K)$ as $n \to \infty$ [4], we optimize for a least-squares solution with bounds $\alpha \in (0.01, 1)$ using `scipy.optimize.least_squares` [8].

The disadvantage of the above general approach compared to the approach presented in the main paper is the direct relationship between the computation cost and $n_r, n_z$, the sizes of supports for $R, Z$.

**Architecture of generative network for radial component**   $G_R$ is a generic multilayer perceptron with layers: (Linear$(1, d_h)$, BatchNorm$(d_h)$, ReLU(), Linear$(d_h, d_h)$, BatchNorm$(d_h)$, ReLU(), Linear$(d_h, 1)$, Exp()), where $d_h$ is the number of nodes in the hidden layers. The number of layers and number of hidden nodes may be modified as needed. Unlike in $G_\mathbf{W}$, the use of batch normalization is not essential in $G_R$.

**Details of Algorithm 3**  To speed up training, we initially resample $\mathcal{Z}$ only once every $k$ mini-batch iterations, decreasing $k$ until $k = 1$ as we approach convergence.

## A.4  Inference and sampling for Archimax copulas

### A.4.1  Inference for Archimax copulas

**Pre-process for extreme-value dependence for initial estimate of stdf**  To determine the block size for the block maximas in (23), we use the test for extreme-value dependence via the max-stable property by Kojadinovic et al. [55], where the max-stable property is defined as:

$$C(u_1, \cdots, u_d) = C^r(u_1^{1/r}, \cdots, u_1^{1/r}), \text{ for } r = 1, 2, \cdots, \mathbf{u} \in [0, 1]^d. \tag{36}$$

The Cramérvon Mises (CvM) distance in (38) between $C(u_1, \cdots, u_d)$ and $C^r(u_1^{1/r}, \cdots, u_1^{1/r})$ is computed using Monte Carlo integration with samples $\mathbf{u}$ drawn uniformly at random from $[0, 1]^d$.

**Details of Algorithm 1 for initial estimate of stdf**  Randomizing the order of observations may help to create different block-maximas in each mini-batch iteration.

**Alternative initialization scheme**  We also considered initialization with different one-parameter families of Archimedean generators, with choice of generator based on the highest log-likelihood of transformed observation $\xi$, from equations (7) and (9). The parameter for each family may be computed from an average of inversion of pairwise Kendall tau, as per the following equation [11], for each pair:

$$\tau_{\varphi,\ell} = \tau_\ell + (1 - \tau_\ell)\tau_\varphi, \tag{37}$$

where $\tau_{\varphi,\ell}$ is the Kendall's tau of the Archimax bivariate marginal, $\tau_\ell$ is the Kendall's tau of the extreme-value component and $\tau_\varphi$ is the Kendall's tau of the Archimedean component. An average of inversion of pairwise Kendall tau was employed in [16], with emphasis on the Clayton generator.

Initialization with specific families of Archimedean generators might bias initialization, and thus we suggested initializing via the Archimedean generator first set to $\varphi(x) = \exp\{-x\}$ representing extreme-value copulas and pre-processing the initial data to have extreme-value dependence via block-maximas. This was also motivated by the experiment on extrapolating to extremes.

**Identifiability**  It follows from a result of Chatelain et al. [16] that the sources of non-identifiability in modeling Archimax copulas are only in: (i) power transformation of $\varphi$ and $\ell$, and (ii) the scale ambiguity of $\varphi$. The power transformation of $\varphi$ and $\ell$ can be illustrated through the following example: Consider both pairs of generators and stdf $(\varphi_a(x) = \exp(-x^{1/\theta})$ and $\ell_a(\mathbf{x}) = \|\mathbf{x}\|_1)$ and $(\varphi_b(x) = \exp(-x)$ and $\ell_b(\mathbf{x}) = \|\mathbf{x}\|_\theta = (x_1^\theta + \cdots + x^\theta)^{1/\theta})$. Both $(\varphi_a, \ell_a)$ and $(\varphi_b, \ell_b)$ lead to the same Archimax copula. The scale ambiguity of $\varphi$ comes from the fact that $\varphi(cx), c > 0$ leads to the same Archimax copula.

We note that these sources of non-identifiability are non-issues in our methods. For the power transformation, there is no ambiguity of $\varphi$ and $\ell$ since the class of $1 - \varphi(1/\cdot)$ where $\varphi(\cdot)$ is calculated as the Williamson $d$-transform of $R$ with a finite support is regularly varying with index $-1$ [16, 5]. In addition, we include a regularization term such that $\mathbb{E}_R[R] = 1$.

### A.4.2  Sampling for Archimax copulas

**Details of Algorithm 4**  Given learned generative networks $G_R, G_\mathbf{W}$, we can generate many samples from the Archimax copula.

## A.5  Background on Archimax copulas

Archimax copulas generalize *Archimedean* and *extreme-value* copulas. They allow asymmetry and arbitrary tail dependence. They were initially developed as a tool to study the behaviour of methods used to estimate the joint distribution of extreme events [10]. The main motivation for Archimax copulas is to model extreme data (e.g. very strong and rare earthquakes) from a mix of moderately

less extreme data (e.g. strong earthquakes) and extreme data. This in turn can be used to generate samples for further studies and simulations.

Archimax copulas were applied to applications such as nutrient intake [71], river flow rates [3] and rainfall [16], where the dependence is asymmetric and sub-asymptotic. In these applications, the authors noted a better fit when using Archimax copulas over Archimedean and extreme-value copulas.

We provide a few connections between Archimax, Archimedean and extreme-value copulas:

- When the *Archimedean generator* $\varphi(x) = \exp(-x)$, and the *radial component* $R \sim$ Erlang$(d)$, Archimax copulas reduce to extreme-value copulas.
- When the *stable tail dependence function (stdf)* $\ell(\mathbf{x}) = (x_1 + \cdots + x_d) = \|\mathbf{x}\|_1$ and the simplex component $\mathbf{S} \sim \text{Unif}(\Delta_{d-1})$, Archimax copulas reduce to Archimedean copulas.

Archimax copulas have intuitive interpretations, such as scale mixture of extremes, dependent frailties and resource sharing. In the case of resource sharing, $R > 0$ is a resource to be distributed randomly among $d$ agents in a way specified by $\mathbf{S}$, where both $R$ and $\mathbf{S}$ are themselves results of independent random processes. For example, $R$ may be profits, and $\mathbf{S}$ may be the way profits is to be divided between stakeholders.

## A.6   Background on multivariate copulas

Copulas are cumulative distribution functions (CDFs) of dependent uniform random variables. They summarize the dependence described by an arbitrary joint CDF after the marginals have been normalized to be uniform. They provide easy marginalization and calculation of tails. In addition, when used in a graphical model, some conditional independence that cannot be easily represented with Markov random fields or Bayesian networks, can be easily represented with cumulative distribution networks [45]. They are also particularly convenient in some applications, such as ranking, where the likelihood is a CDF [43].

For an introduction to copulas, the following textbooks and collection of works are great resources [32, 74, 51, 49, 50].

We also summarize the common multivariate copulas in Table 5.

Table 5: Multivariate copulas

| | |
|---|---|
| GAUSSIAN (GC) | $\Phi_{\mathbf{R}}(\Phi^{-1}(u_1), \cdots, \Phi^{-1}(u_d))$ |
| VINE (RV, CV, DV) | $\prod_{e \in \mathcal{E}(\mathcal{V})} c_{U_{e_1}, U_{e_2} \mid \{U_{e_d}\}}(F_{U_{e_1} \mid \{U_{e_d}\}}(u_{e_1}), F_{U_{e_2} \mid \{U_{e_d}\}}(u_{e_2}))$ |
| ARCHIMEDEAN (AC) | $\varphi(\varphi^{-1}(u_1) + \cdots + \varphi^{-1}(u_d))$ |
| H. ARCHIMEDEAN (HAC) | $C_0(C_1(\cdots), \cdots, C_J(\cdots)), C_i \in \text{AC}$ |
| EXTREME-VALUE (EV) | $\exp\{-\ell(-\log(u_1), \cdots, -\log(u_d))\}$ |

The Gaussian copula (GC) has a tractable expression for both the CDF and the density. However, it is independent in the tails, a significant reason why Gaussian copula is not suitable for modeling financial risks. Vine copulas, such as R-vines (RV), C-vines (CV) and D-vines (DV), are computationally intensive and hard to interpret due to repeated conditioning with pair copulas. Archimedean copulas are symmetric in all coordinates, which is an assumption that is usually not held in practice. Hierarchical Archimedean copulas aim to break this symmetry but are difficult to construct due to nesting conditions that are hard to satisfy. Extreme-value copulas are max-stable copulas which results in lower tail independence, an assumption that is sometimes not held in practice. Many copulas are not flexible in the extremes, which leads to independence, except in the case of Archimdean copulas which only Gumbel copulas satisfy the tail dependence. In high dimensions, none of the existing copulas typically fit data well. Model misspecification is often accepted in return for tractability, and some dependence is better than independence [41].

Inferring the parameters of a copula is usually done via maximum likelihood estimation if a density can be computed, or by using minimum distance estimator and goodness-of-fit tests if a density cannot be computed. In both cases, expectations are usually replaced by their empirical versions. For more background on estimating copulas, see [15].

Sampling from a copula using the conditional sampling method with Rosenblatt transform is usually not possible in high dimensions, due to repeated differentiation. In our experiments, the conditional sampling method, using automatic differentiation in `PyTorch` breaks down at dimension $d = 4$. In general, only models with stochastic representation may be easy to sample [69].

### A.7 Background on copulas in machine learning

Copulas is a rising topic in machine learning, as evident from numerous publications, including but not limited to:

- Cumulative distribution networks, modeled as a product of copulas [45, 44, 46]. They can represent some conditional independencies not represented by Markov random fields and Bayesian networks, allow loops [52] and mixed graphs [85], with application to ranking [43] and heavy-tailed distributions [52].
- Copula Bayesian networks, modeled as a product of conditional copulas [26], with application to missing data [27], classification [28], time-series [25] and fast structure learning [89, 90].
- Copula processes [98] and application of copulas to time-series [66, 39, 83, 95].
- Copula based dependence measures and distances [67, 57, 80, 13, 70].
- Copula variational inference, allowing dependencies between latent variables [91, 37, 40].
- Generative modeling [88, 17, 59, 48, 1, 7].
- Applications of copula in areas including graph neural networks [68], multi-label learning [64], multi-agent interactions [93], bundle pricing [61], missing value [94, 58, 100], sparse representation [96], outlier detection [62], causal discovery [20], domain adaptation and transfer learning [65, 84] and structure learning [12].
- Recent work on deep network based copulas, including Archimedean [63, 75], extreme-value [38], autoregressive [76, 53] and transformer-attentional copulas [24].

## B  Experiments

The metric we use to compare the methods is based on the Cramér-von Mises (CvM) statistic [81] which is defined as:

$$\mathbf{CvM} = \int (C_{*,n}(\mathbf{u}) - C_{\theta,n}(\mathbf{u}))^2 \, \mathrm{d}\mathbf{u}, \tag{38}$$

where $C_{*,n}$ is the empirical copula of true samples and $C_{\theta,n}$ is the empirical copula of generated samples and the integral is computed using Monte Carlo integration with 10,000 samples of $\mathbf{u}$ drawn uniformly at random from $[0, 1]^{d-1}$.

The empirical copula for $n$ given observations $(u_{11}, \cdots, u_{1d}), \cdots, (u_{n1}, \cdots, u_{nd})$ is defined as:

$$C_n(\mathbf{u}) = \frac{1}{n} \sum_{i=1}^{n} \mathbb{1}\{u_{i1} \leq u_1, \cdots, u_{id} \leq u_d\}. \tag{39}$$

All timings are with a 2.7 GHz Intel Core i7, 16GB 2133MHz LPDDR3.

### B.1  Inference for Archimedean generator

We summarize the common Archimedean generators in Table 6.

The Clayton (C) generator is lower tail dependent, upper tail independent, the Frank (F) generator is symmetric in both lower and upper tails, the Joe (J) and Gumbel (G) generators are lower tail independent and upper tail dependent. Thus the above generators represent different radial envelopes.

The map $\lambda$ is commonly used to estimate and evaluate estimates of $\varphi$ [33, 34]. The map $\lambda$ is defined as:

$$\lambda(w) = \varphi^{-1}(w)/(\varphi^{-1}(w))' = \{\varphi' \circ \varphi^{-1}(w)\}\varphi^{-1}(w), \tag{40}$$

Table 6: Archimedean generators

| | $\varphi_\theta(x)$ | $\theta_{\tau=0.2}$ | $\theta_{\tau=0.5}$ |
|---|---|---|---|
| CLAYTON (C) | $(1+x)^{-1/\theta}$ | 0.5 | 2 |
| FRANK (F) | $-\log(1 - (1 - \exp(-\theta))\exp(-x))/\theta$ | 1.86 | 5.74 |
| JOE (J) | $1 - (1 - \exp(-t))^{1/\theta}$ | 1.44 | 2.86 |
| GUMBEL (G) | $\exp(-t^{1/\theta})$ | 1.25 | 2 |

such that an estimate of $\varphi$ can be recovered from $\lambda$ as

$$\varphi^{-1}(w) = \exp\left\{\int_{w_0}^w 1/\lambda(t)dt\right\}. \tag{41}$$

It is more convenient to present results in terms of $\lambda$ since it is scale invariant, unlike $\varphi$, where $\varphi(cx)$ for any $c > 0$ lead to the same copula. In dimension $d = 2$, $\lambda$ is directly related to the Kendall distribution function as $\lambda(w) = w - K(w)$. As such, the asymptotic variance for $\lambda_n$ may be computed from the asymptotic variance for $K_n$, where $n$ is the number of observations. This relationship is more complicated in dimension $d > 2$ but often used as an approximate, useful for drawing confidence bands around $\lambda_n$ to reject models whose $\lambda_\theta$ fall outside the band. In addition, the asymptotic variance of the independence copula may be easily computed as:

$$\sigma_{\lambda_n}^2(x) = \frac{x(x - \log(x) - 1)}{n}. \tag{42}$$

**Selection of support sizes** $n_r, n_z$     For $n_r = 100$ and $n_z \in \{20, 30, \cdots, 100\}$, we plot the estimates of $\lambda$ in Figure 3. Results from different runs are in blue. The ground truth is a Clayton generator with $\tau = 0.2$ and a negative scaled extremal Dirichlet stdf with $\alpha = (1, 1, 1, 1, 2, 2, 2, 3, 3, 4), \rho = 0.69$. The ground truth is plotted in black and approximate confidence bands around the ground truth is in dotted black. The computed the mean squared error (MSE) in fitting $K$ and $\lambda$ and the time taken are given in Figure 4 and Table 7, where the standard deviation is given in parenthesis.

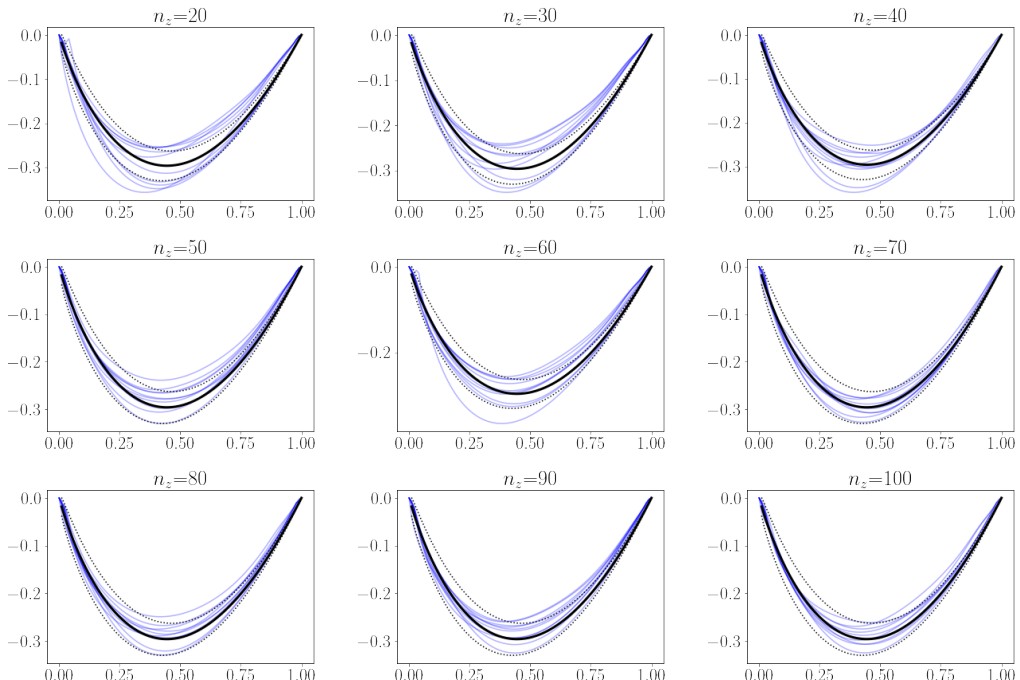

Figure 3: Estimates of $\lambda$ with $n_r = 100$ and $n_z \in \{20, \cdots, 100\}$. Ground truth in black, approximate confidence bands in dotted black. Estimates from different runs are in blue.

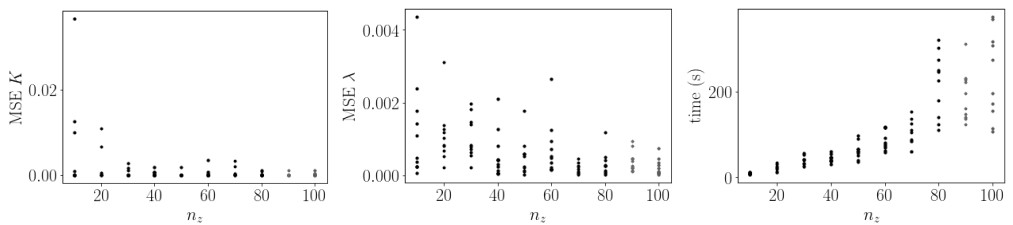

Figure 4: Mean squared error (MSE) in fitting $K$ and $\lambda$ and the time taken.

Table 7: Mean squared error (MSE) in fitting $K$ and $\lambda$ and the time taken.

| $n_r = 100, n_z$ | 10 | 20 | 30 | 40 | 50 | 60 | 70 | 80 | 90 | 100 |
|---|---|---|---|---|---|---|---|---|---|---|
| MSE $K \times 10^{-3}$ | 5.99(11.08) | 1.84(3.60) | 0.55(0.91) | 0.34(0.56) | 0.21(0.57) | 0.46(1.02) | 0.60(1.06) | 0.23(0.39) | **0.12(0.31)** | 0.16(0.31) |
| MSE $\lambda \times 10^{-3}$ | 1.24(1.27) | 1.10(0.75) | 1.04(0.56) | 0.57(0.62) | 0.49(0.49) | 0.74(0.72) | **0.18(0.13)** | 0.32(0.32) | 0.39(0.27) | 0.23(0.21) |
| TIME (SEC) | 8.59(1.78) | 21.30(7.10) | 40.06(11.11) | 43.46(8.23) | 62.46(18.45) | 86.08(21.91) | 100.62(29.75) | 217.68(71.45) | 190.10(56.29) | 239.04(96.97) |

From the results, for $n_r = 100$, it would be appropriate to use $n_z \in \{70, \cdots, 100\}$, with a tradeoff between accuracy and computation time.

**Plots of $\lambda$ for sample size** $n = 1000$ While our method infers an arbitrary Archimedean generator and takes the joint dependence across covariates into account, the method in [16] infers a Clayton generator from pairwise Kendall taus. Thus the performance gap between our method and the method in [16] is expected to increase as the generator differs from the Clayton generator and as the observations become less symmetric.

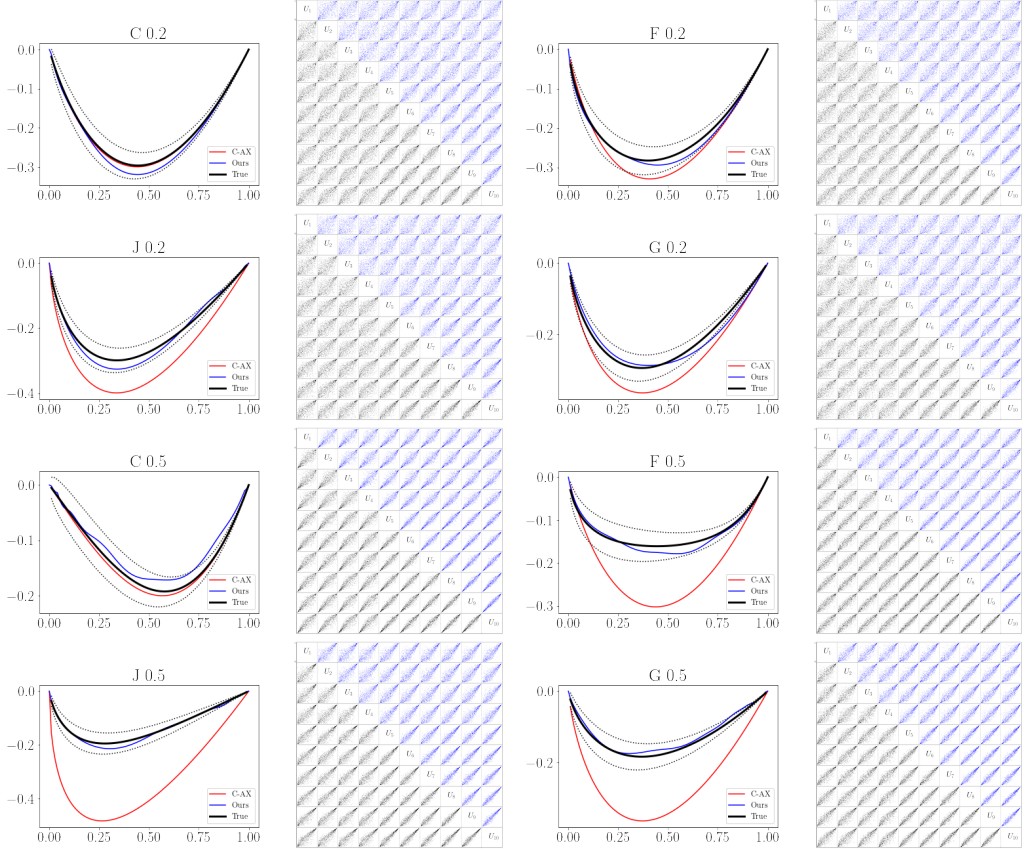

Figure 5: Estimates of $\lambda$ given samples of **S** from true $\ell$ for $n = 1000$. Ground truth in black, approximate confidence bands in dotted black, the method from [16] in red, our method in blue. On the right of each plot of $\lambda$ is a plot of samples from the copula, ground truth below the diagonal in black, our method above the diagonal in blue.

**Plots of $\lambda$ for sample size** $n = 200$    Increasing the number of observations $n$ improved estimation accuracy, hinting at consistency.

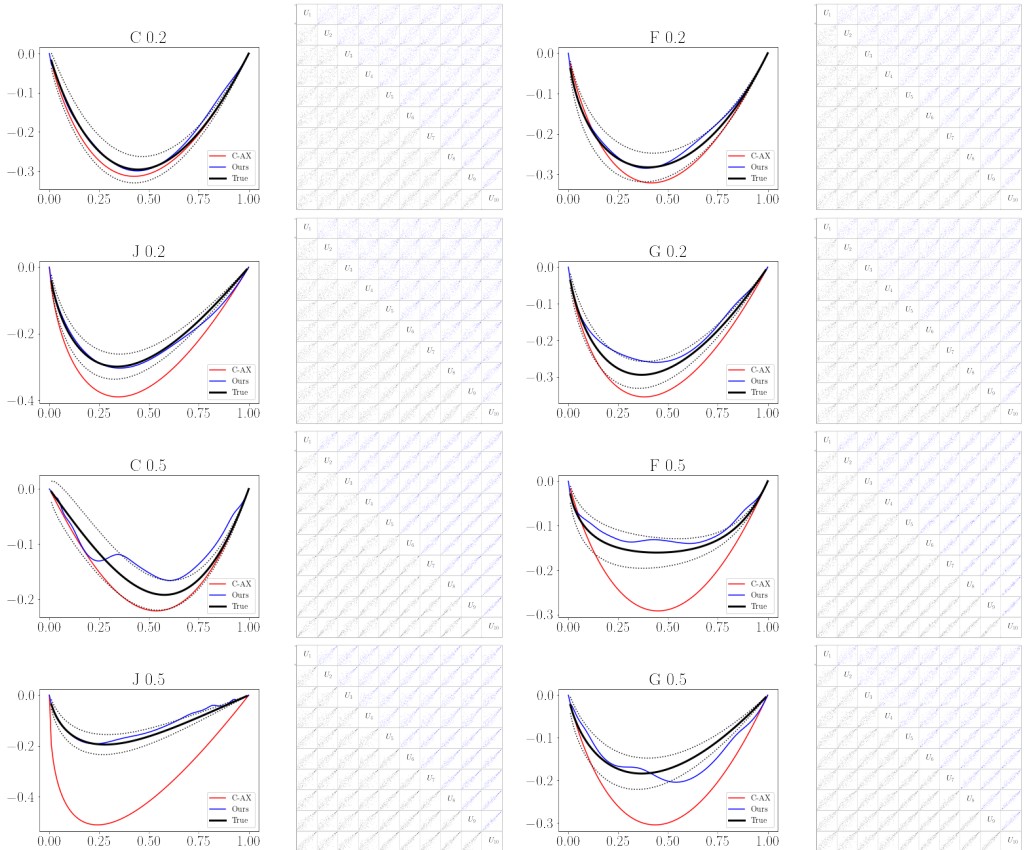

Figure 6: Estimates of $\lambda$ given samples of **S** from true $\ell$ for $n = 1000$. Ground truth in black, approximate confidence bands in dotted black, the method from [16] in red, our method in blue. On the right of each plot of $\lambda$ is a plot of samples from the copula, ground truth below the diagonal in black, our method above the diagonal in blue.

## B.2 Inference for stable tail dependence function and sampling for simplex component

The negative scaled extremal Dirichlet (NSD) [5] is a rich class encompassing many parametric models of the stdf and spectral component, including the logistic, asymmetric logistic, negative logistic, and extremal Dirichlet models [18].

It is specified by:

$$\ell(\mathbf{x}) = \frac{\Gamma(\alpha_1 + \cdots + \alpha_d - \rho)}{\Gamma(\alpha_1 + \cdots + \alpha_d)} \mathbb{E}_{\mathbf{D}} \left[ \max_{j=1,\cdots,d} \left( \frac{x_j D_j^{-\rho} \Gamma(\alpha_j)}{\Gamma(\alpha_j - \rho)} \right) \right], \tag{43}$$

where $\mathbf{D} = (D_1, ..., D_d)$ is distributed as a Dirichlet$(\alpha_1, ..., \alpha_d)$ with $\alpha_1, ..., \alpha_d > 0$ and $\rho \in (0, \min(\alpha_1, \cdots, \alpha_d))$.

The integrated relative absolute error (IRAE) is commonly used to evaluate estimates of $\ell$, and is given by [16]:

$$\mathrm{IRAE}(\ell, \ell_\theta) = \frac{1}{|\Delta_{d-1}|} \int_{\Delta_{d-1}} |\ell(\mathbf{x}) - \ell_\theta(\mathbf{x})|/\ell(\mathbf{x}) \, d\mathbf{x}. \tag{44}$$

The IRAE is computed using Monte Carlo integration with 10,000 samples $\mathbf{x}$ drawn uniformly at random from the simplex $\Delta_{d-1}$.

For given true $\varphi$, we report the IRAE of the modified Pickands estimator from [16], the modified CFG estimator from [16], and our method in Table 8. The stdf is a NSD with $\alpha = (1, 1, 1, 1, 2, 2, 2, 3, 3, 4)$, $\rho = 0.69$.

Table 8: Inference of $\ell$ given true $\varphi$

|  | C 0.2 | C 0.5 | F 0.2 | F 0.5 | J 0.2 | J 0.5 | G 0.2 | G 0.5 |
|---|---|---|---|---|---|---|---|---|
| IRAE $\pm 0.01$ P [16] | 0.16 | 1.00 | 0.05 | 0.06 | 0.06 | 0.07 | 0.08 | 0.17 |
| IRAE $\pm 0.01$ CFG [16] | 0.05 | 0.11 | 0.04 | 0.04 | 0.05 | 1.00 | 0.06 | 0.15 |
| IRAE $\pm 0.01$ (OURS) | 0.06 | 0.12 | 0.04 | 0.05 | 0.06 | **0.07** | 0.08 | 0.15 |

As mentioned in Appendix A.2.1, though our estimator is based on the Pickands estimator, it performs better than the Pickands estimator. This suggests that our direct method of training with the generative network generating the class of valid stdfs would perform better than the alternative method that first estimates $\hat{\ell}$ then train the generative network to match $\hat{\ell}$. As noted in [16], the modified CFG estimator performs better than the modified Pickands estimator for Archimax copulas. A future direction may be to improve the robustness of our estimator with the CFG modification.

We also provide the IRAE and time taken versus number of minibatch iterations in Figure 7. The plots suggest that our algorithm converges and is not computationally intensive.

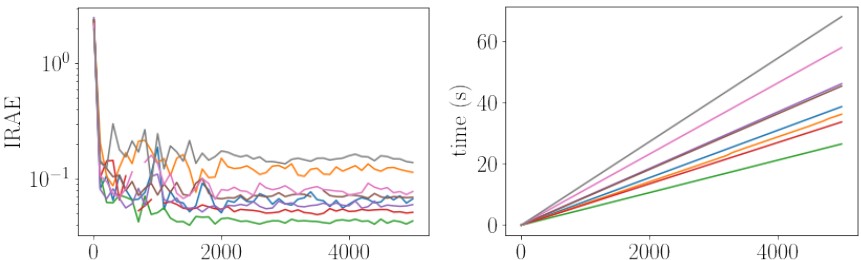

Figure 7: IRAE in fitting $\ell$ and the time taken versus number of mini-batch iterations. Each line is for a different copula setting in Table 8.

### B.3 Modeling nutrient intake

The data and documentation from the study of nutrient intake in women is made available by the U.S. Department of Agriculture (USDA) [92]. There are $n = 1459$ observations of dimension $d = 17$, corresponding to the variables: Energy, Protein, Vitamin A (IU), Vitamin A (RE), Vitamin E, Vitamin C, Thiamin, Riboflavin, Niacin, Vitamin B6, Folate, Vitamin B12, Calcium, Phosphorus, Magnesium, Iron, Zinc. This dataset was previously studied in [71] where using a Clayton-NSD Archimax copula improved fit over a Clayton Archimedean copula. However, in [71], the experiment was limited to only $n = 737$ and $d = 3$, corresponding to Calcium, Iron and Protein.

We compared our method to copula based models and deep network based models from literature using the CvM distance in (38). The abbreviations for the copula based models in Table 2 of the main paper correspond to the abbreviations in Table 5 summarizing the common multivariate copulas. They are Gaussian (GC), R-vine (RV), C-vine (CV), D-vine (DV), Archimedean (AC *), hierarchical Archimedean (HAC) [36] and extreme-value (EV †) copulas. The abbreviation (C-AX †) corresponds to the state-of-the-art in inferring Archimax copulas with a Clayton generator for $\varphi$ and the modified CFG estimator for $\ell$ [16]. The abbreviations for the deep network based models in Table 2 are Wasserstein GAN with gradient penalty (WGAN), masked autoregressive flow (MAF) and variation autoencoders (VAE). Lastly, the abbreviation (Gen-AX *†) corresponds to our method. For methods marked with *, we use $\varphi$ described in Algorithm 3 and for methods marked with † we use $\ell$ and the sampling methods described in Algorithms 2 and 4.

Table 2 shows our method outperforming the above methods using the CvM distance.

We additionally provide plots of generated samples versus true samples in Figure 9 (a-l). The generated samples are plotted in blue above the diagonal while the true samples are plotted in black below the diagonal. From the plot of our generated samples in Figure 9 (l), an improvement can be made with hierarchical Archimax copulas [42] to have different Archimedean generators $\varphi$ and thus different radial envelopes for covariates.

We briefly describe the training process using our proposed method. We first initialize $\ell_\theta$ with Algorithm 1 on block-maximas. Using the test for extreme-value dependence [55], we chose the block size $n/k = 5$. The plots for block sizes $n/k \in \{1, 2, 5, 10\}$ and exponents $r \in \{2, 3, \cdots, 10\}$ are given in Figure 8.

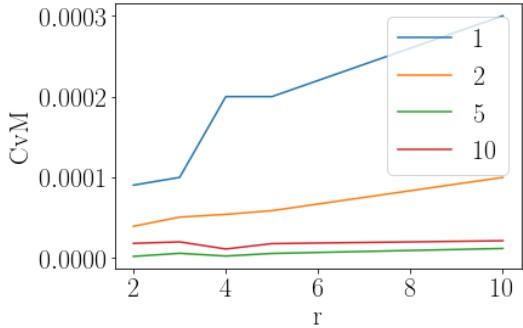

Figure 8: Selection of block size $n/k = 5$ using the test for extreme-value dependence [55].

We compared our initial estimate $\ell_\theta$ to the state-of-the-art CFG estimator applied on the block maximas $\hat{\ell}$ [10]. The plot of IRAE against mini-batch iterations show convergence in 2000 mini-batch iterations, with a duration of 20s, and an IRAE of 0.08. We then learn $\varphi_\theta$ with Algorithm 3 on the full dataset, with samples of $\mathbf{S}$ from Algorithm 2. The plot of MSE in fitting the empirical Kendall distribution show convergence in 2000 mini-batch iterations, with a duration of 25s, and an MSE of 0.0008. We note that the initialization scheme seems to be performing well since the learnt $\varphi_\theta$ focused on modifying only the lower tail. We then update $\ell_\theta$ with Algorithm 1 on the full dataset, given $\varphi_\theta$. The NLL of transformed observations $\xi$ went from 1.67 in the initialization to -1.12 with the use of $\varphi_\theta$. The IRAE to the state-of-the-art modified CFG estimator on the full dataset was 0.078.

The estimation of the copula based models was done using the `Copulas` library [21] in Python and the `HACopula` toolbox [35] in MATLAB.

The architectures of the deep networks are:

- WGAN: 3 layers for generator, 3 layers for discriminator, hidden size 128.
- MAF: 2 flows, hidden size 128 for each flow.
- VAE: 3 layers for encoder, 2 layers for decoder, hidden size 128, latent size 16.
- Gen-AX: 3 layers hidden size 30 for $G_{\mathbf{W}}$, 3 layers hidden size 10 for $G_R$.

The implementation was with `PyTorch`, the Adam optimizer was used with learning rate 1e-3.

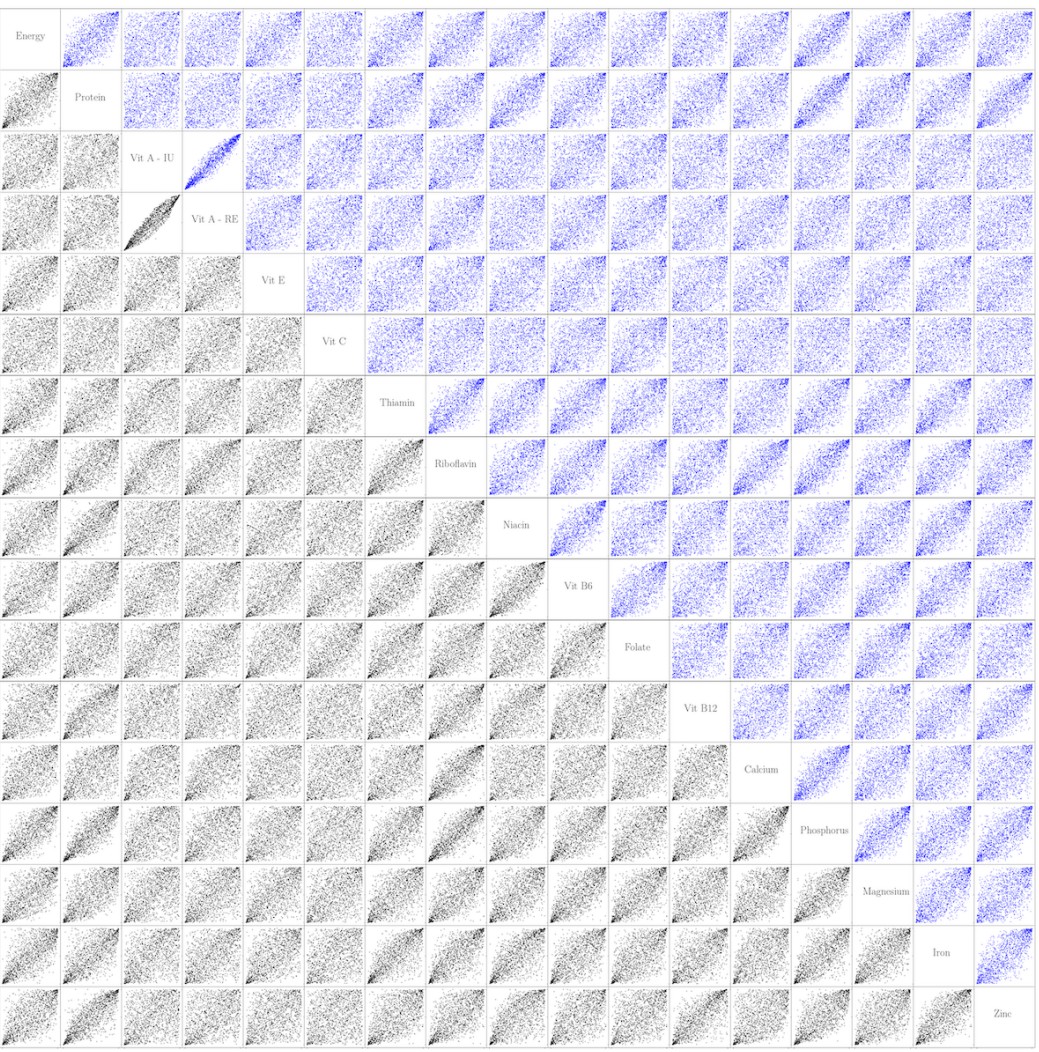

Figure 9: (a) Gaussian copula (GC).

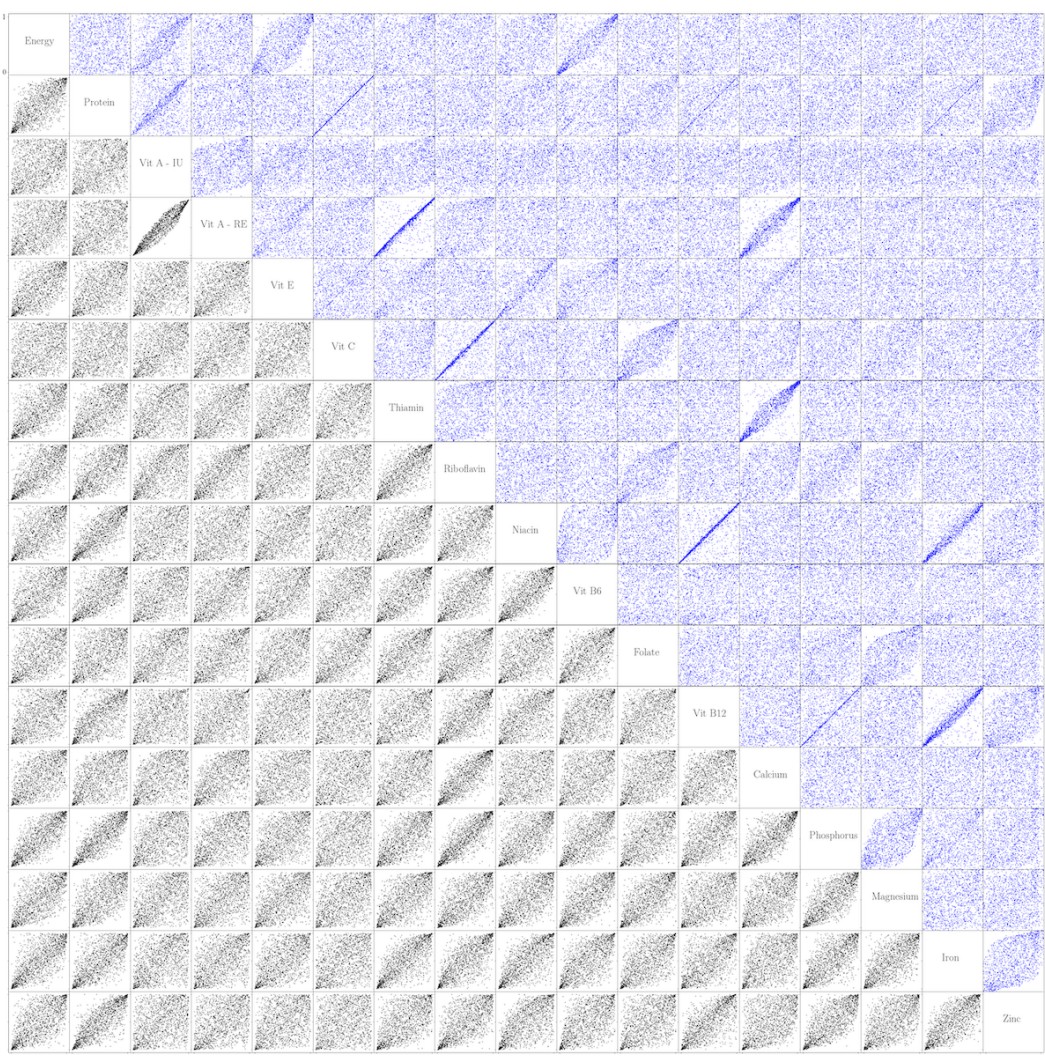

Figure 9: (b) R-vine (RV).

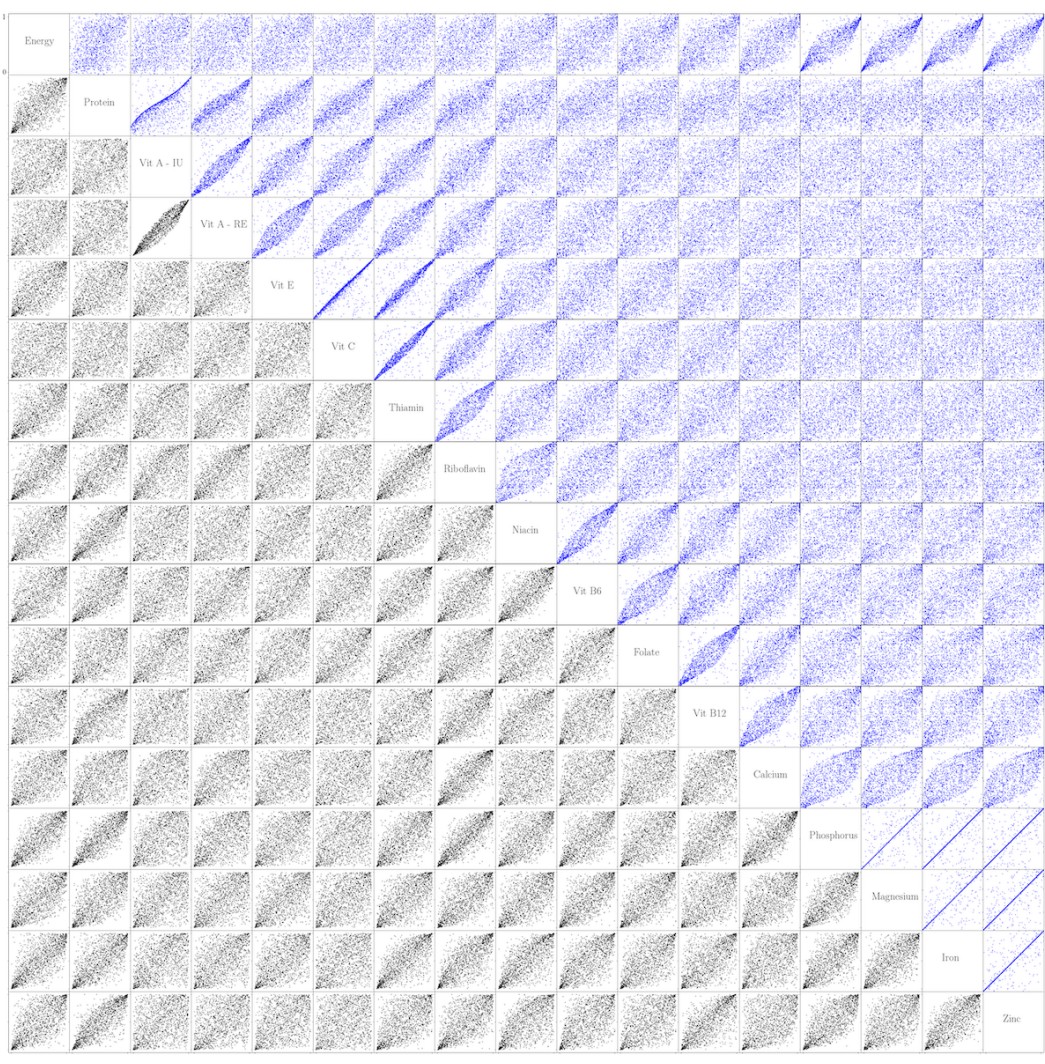

Figure 9: (c) C-vine (CV).

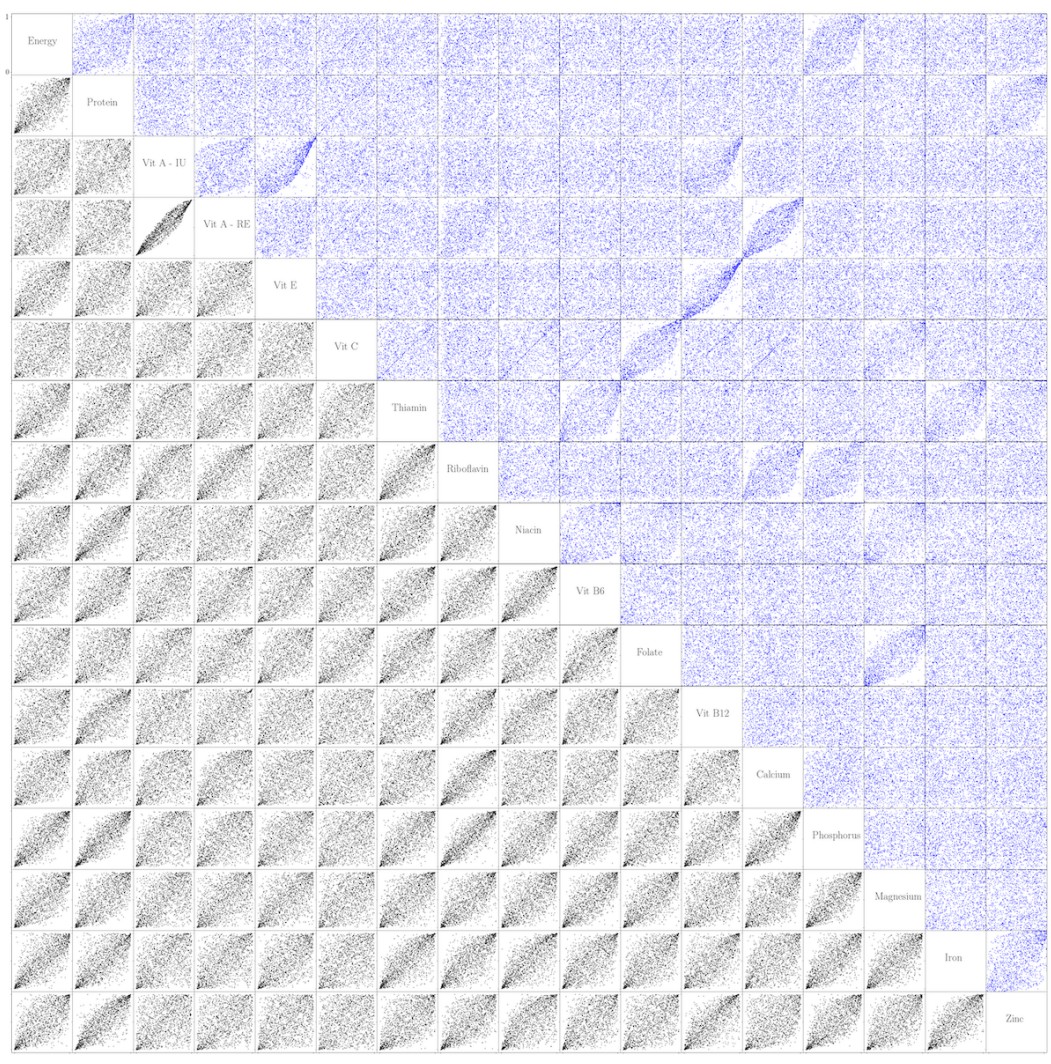

Figure 9: (d) D-vine (DV).

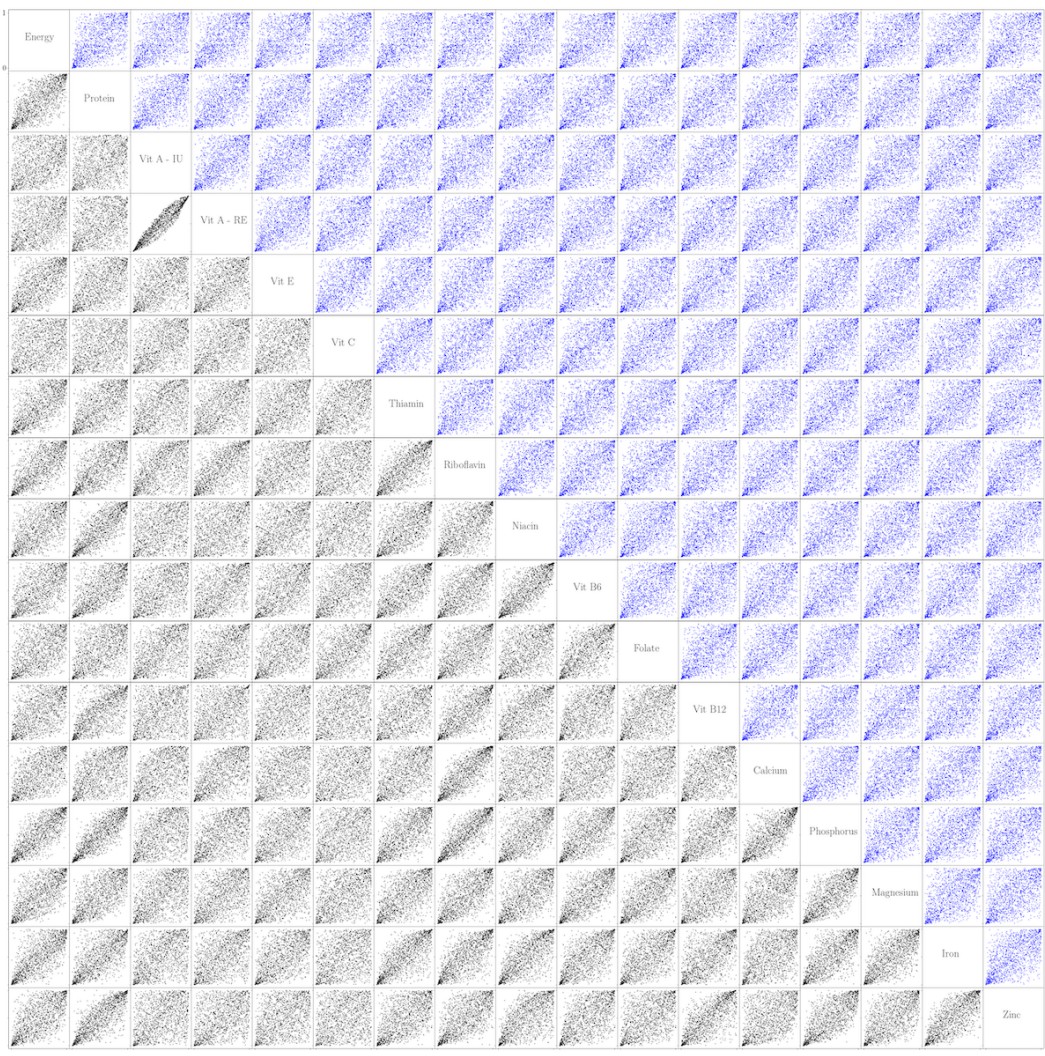

Figure 9: (e) Archimedean copula (AC *) inferred with Algorithm 3.

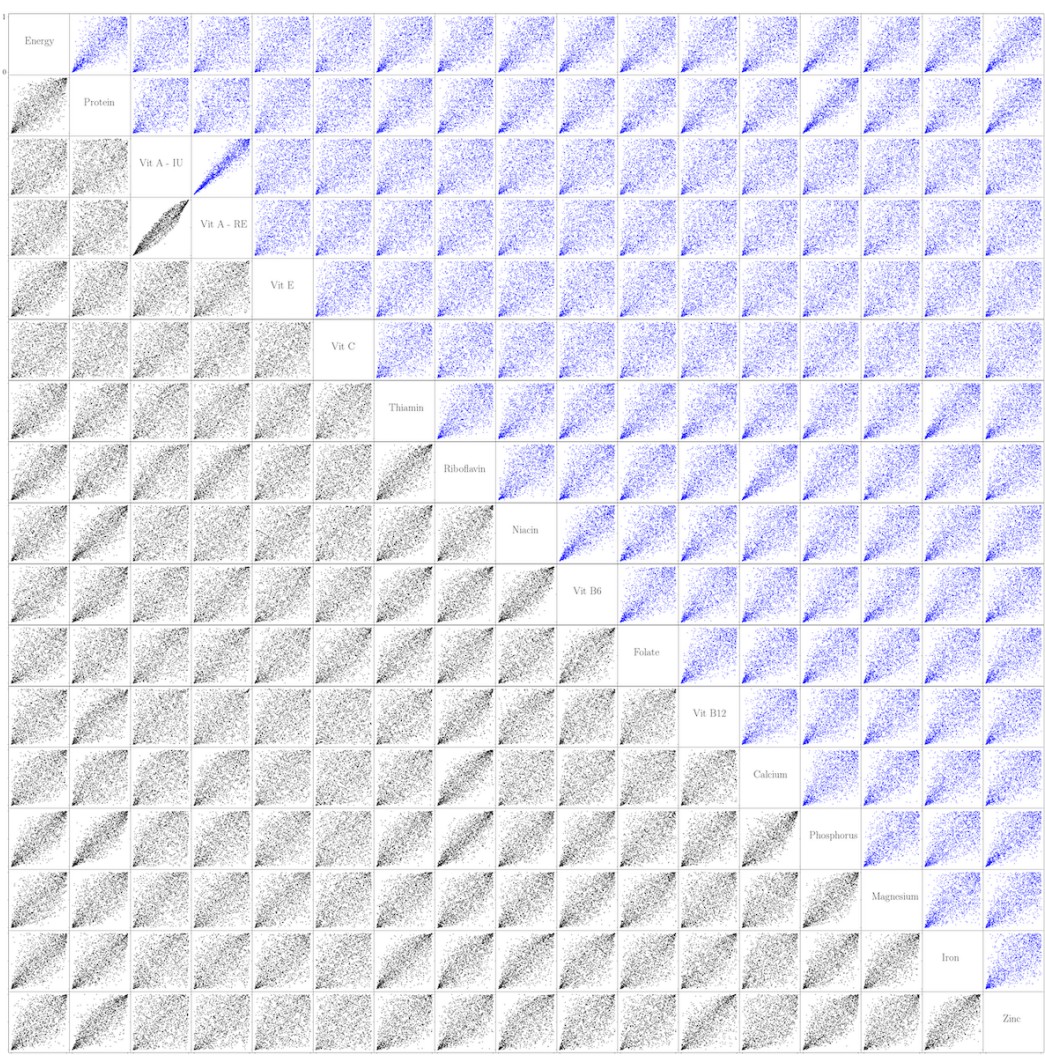

Figure 9: (f) Hierarchical Archimedean copula (HAC) [36].

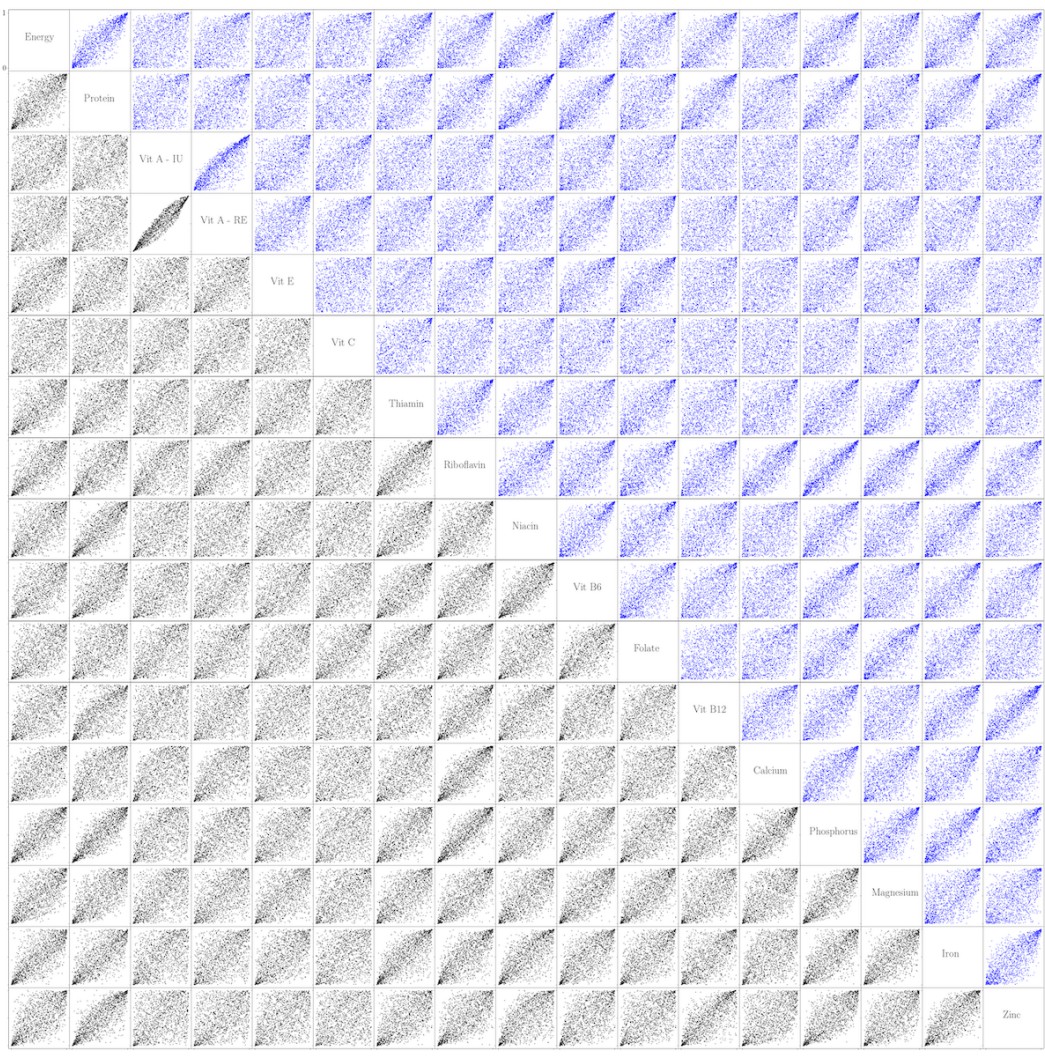

Figure 9: (g) Extreme-value copula (EV †) sampled with Algorithms 2 and 4.

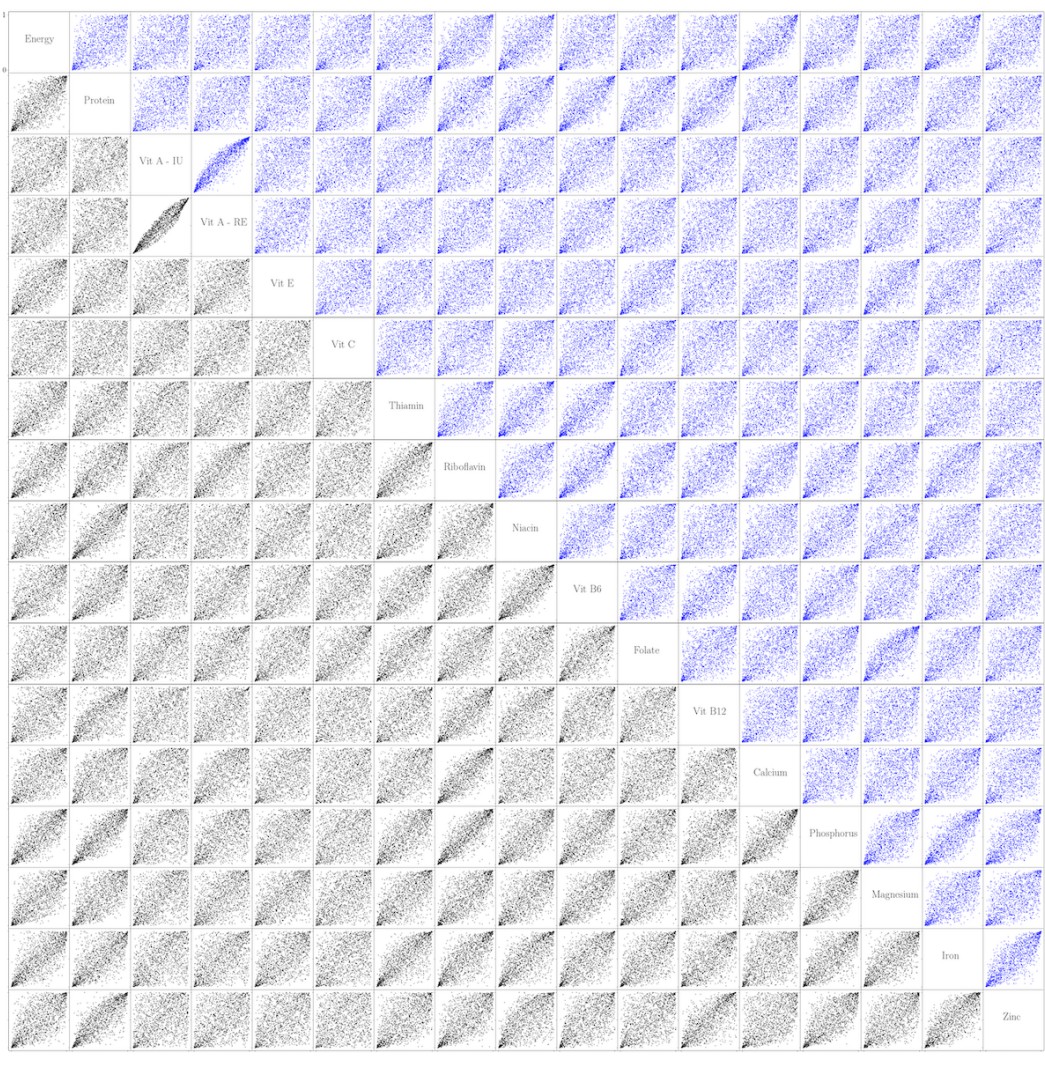

Figure 9: (h) Archimax copula (C-AX †) inferred with the state of the art [16], sampled with Algorithms 2 and 4.

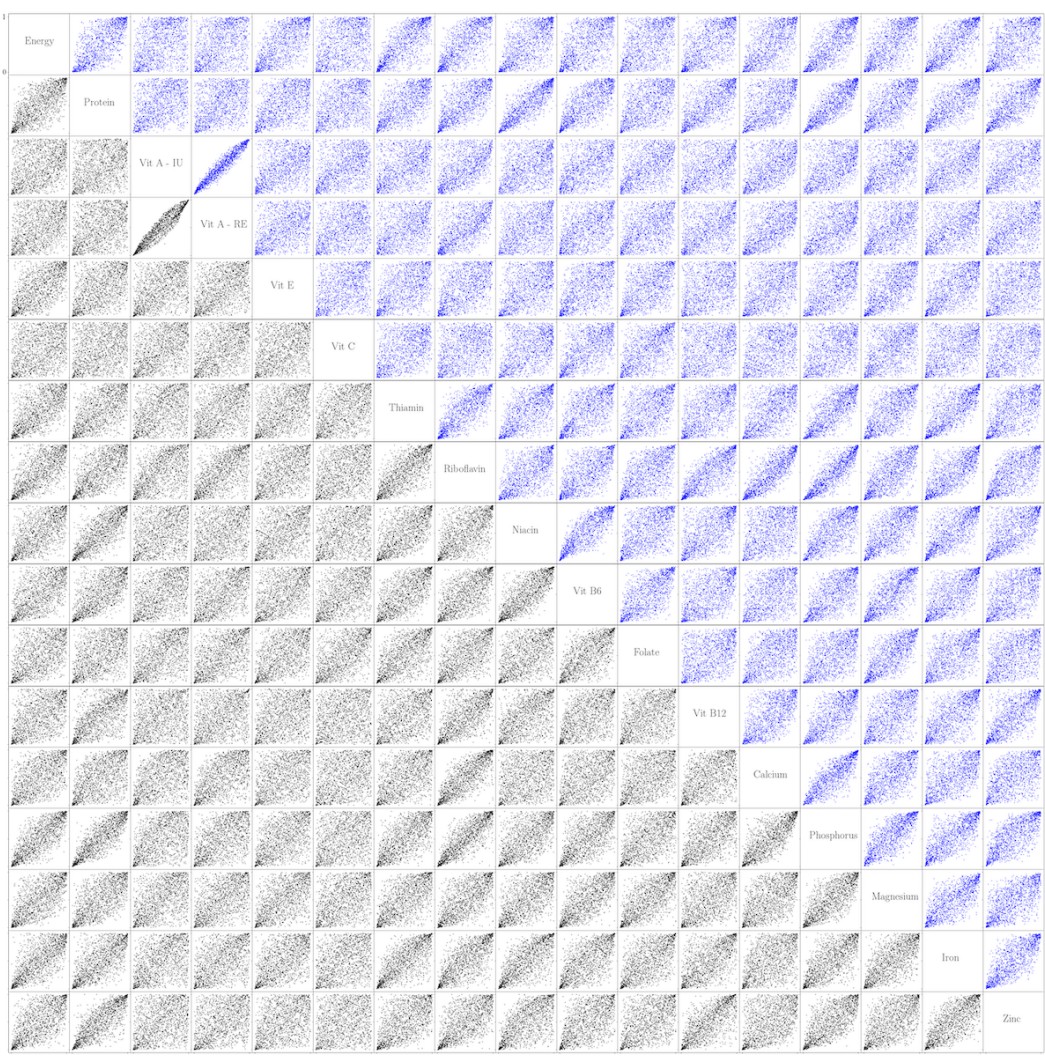

Figure 9: (i) Wasserstein GAN with gradient penalty (WGAN).

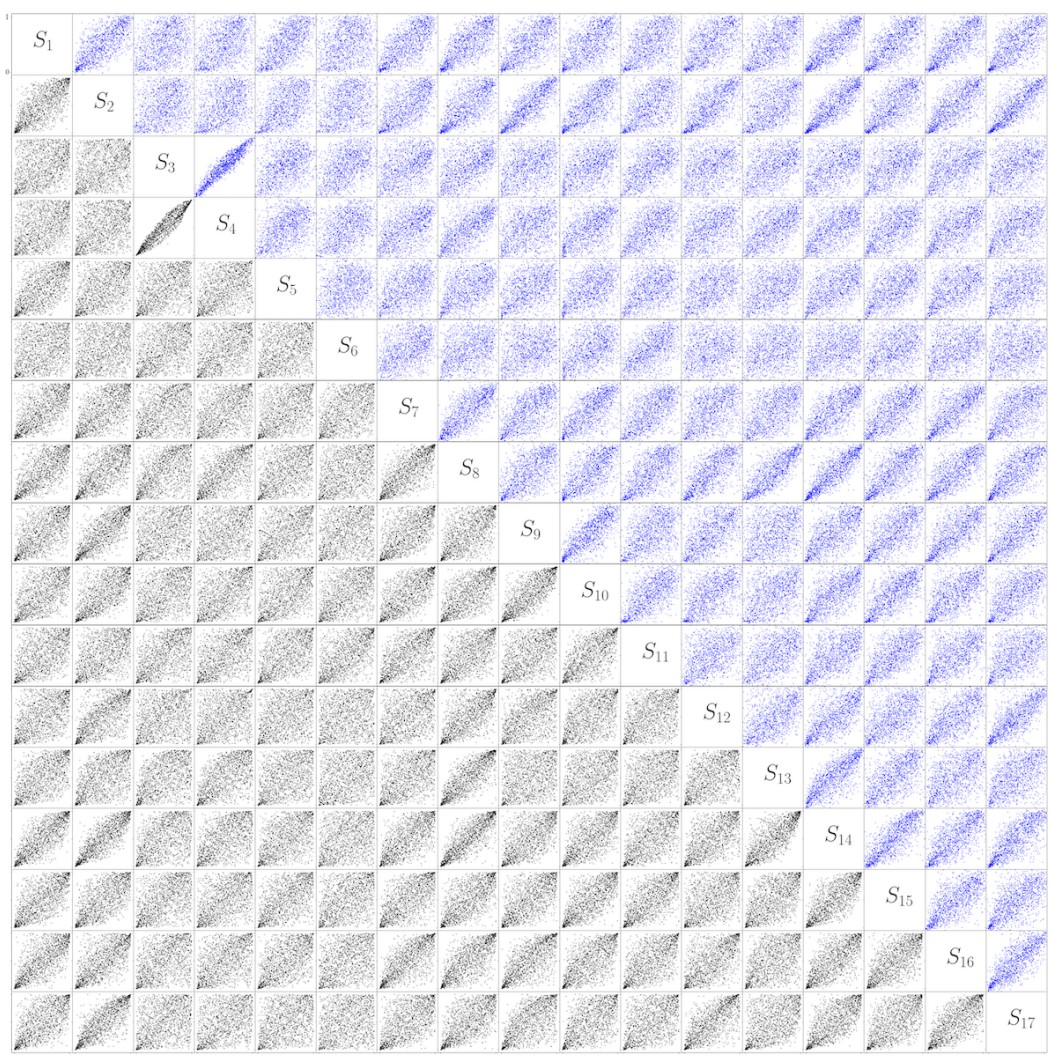

Figure 9: (j) Masked autoregressive flow (MAF).

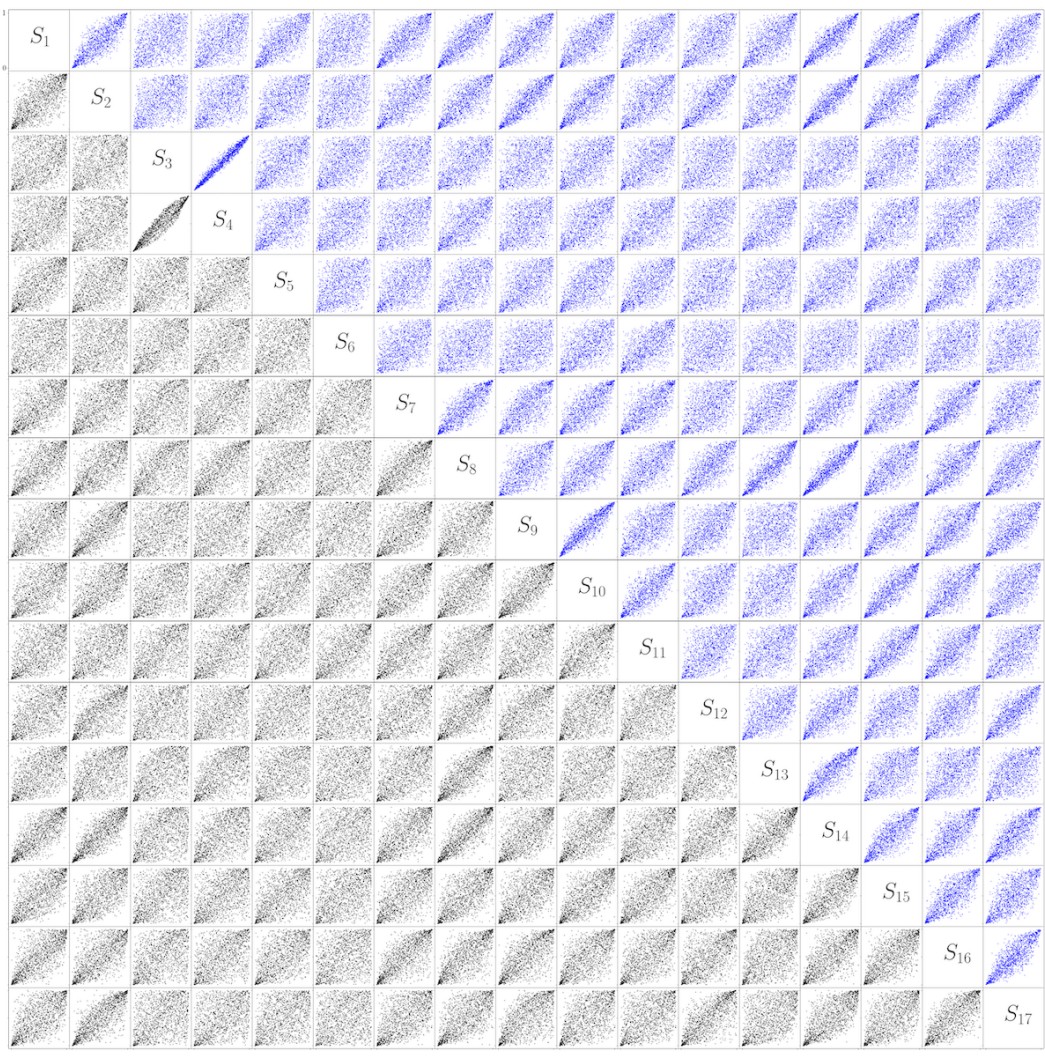

Figure 9: (k) Variational autoencoder (VAE).

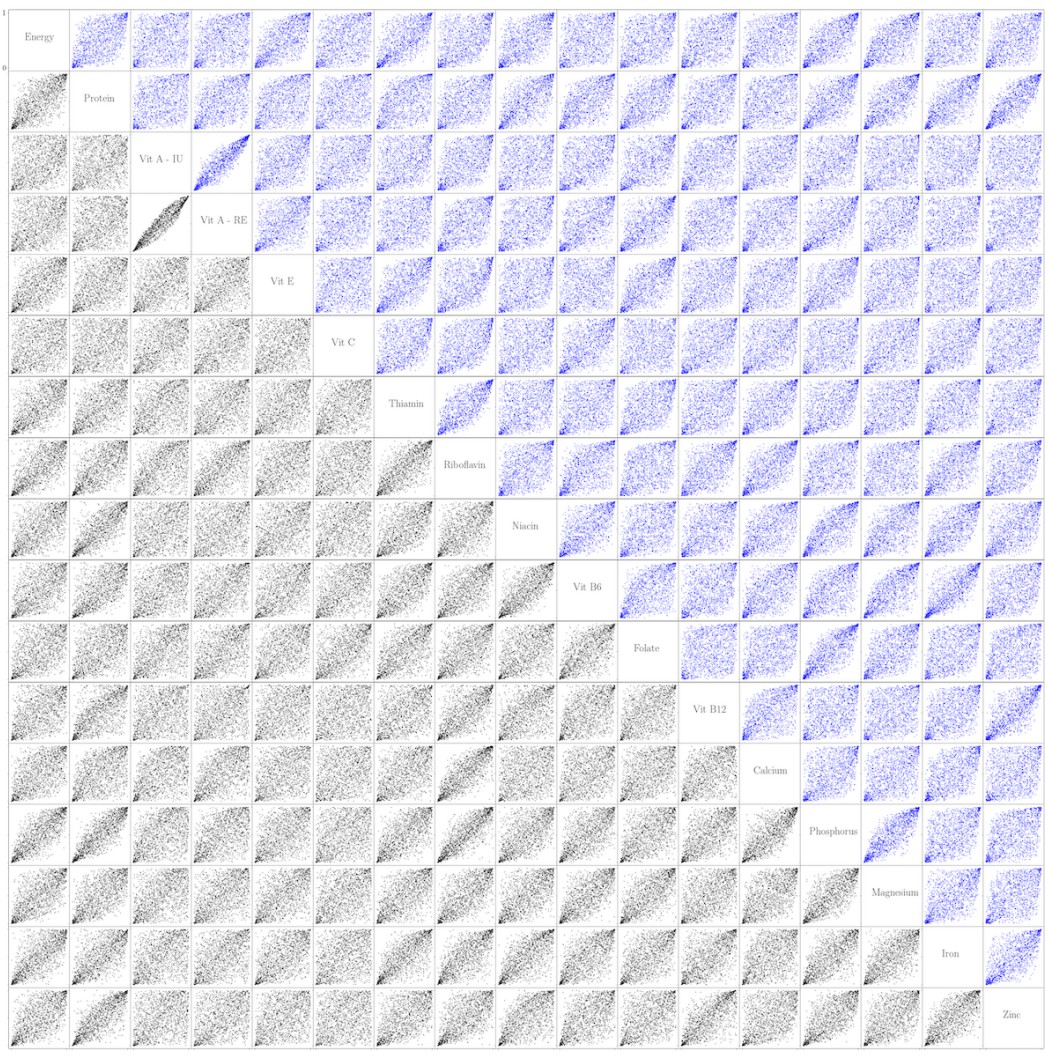

Figure 9: (l) Archimax copula (Gen-AX *†) inferred and sampled with our methods.

## B.4 Extrapolating to extreme rainfall

This data simulates the monthly rainfall for 3 locations (Belle-Ile, Groix, and Lorient) in French Brittany, where the dependence is asymmetric and non-extreme. We follow the Archimax model in [16] and extend the experiment with different Archimedean generators of the same Kendall tau $\tau$. The methods used for comparison are Wasserstein GAN with gradient penalty (WGAN), masked autoregressive flow (MAF), variational autoencoder (VAE) and the state-of-the-art Clayton-Archimax copula (C-AX). All methods were first trained on all observations $n = 240$ with dimension $d = 3$. Many samples were then generated from the trained model to estimate $\hat{\ell}$ from block maximas using the state-of-the-art CFG estimator for extreme-value copulas [10]. As mentioned in the main paper, for extrapolating to extremes, we did not compare to other copula based models as many classical copulas are independent or Gumbel in the extremes.

Table 3 of the main paper show our method performs consistently across the different Archimedean generators. In addition, our method always performs the best and in the case of ties, always among the best. Plots of the generated samples and generated samples in the extremes are given in Figure 10, with the Clayton-NSD Archimax copula experiment setting.

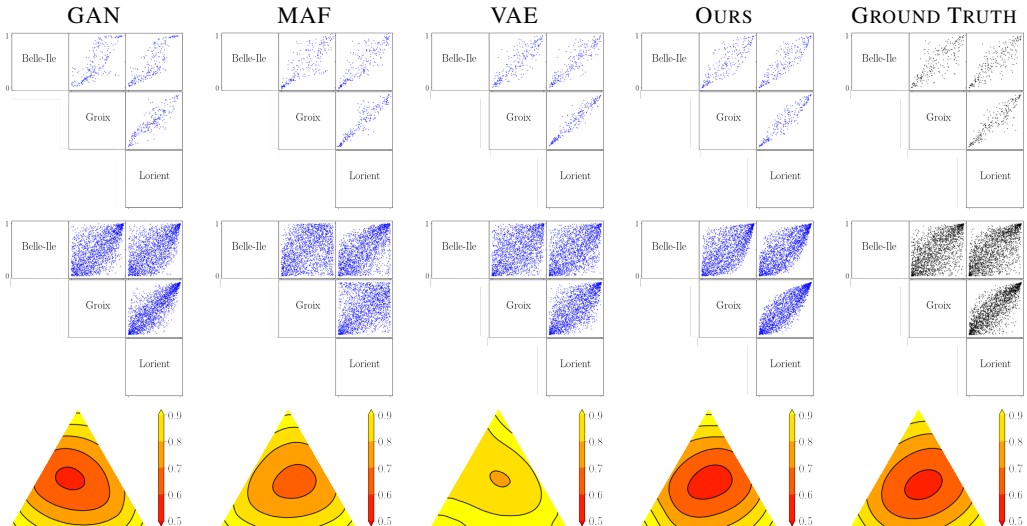

Figure 10: Extrapolating to dependence in the extremes. Plots of generated samples (top), generated samples in the extremes (middle) and $\ell(\mathbf{w}), \mathbf{w} \in \Delta_{3-1}$ (bottom).

### B.4.1 Modeling monthly rainfall

Although we did not compare to other copula based models for extrapolating to extremes, we compared to other copula based models for modeling monthly rainfall. In particular, the skew-t copula is a good addition to one's arsenal for flexible asymmetrical copulas [23, 56, 99, 87].

For the Clayton-NSD Archimax copula experiment setting, the CvM distance for Archimax, Gaussian, extreme-value, t and skew-t copulas are: **0.0003** (lower is better), 0.0005, 0.0006, 0.0008, 0.0027.

For this scenario, the Archimax and Gaussian copula performed better than the extreme-value, t and skew-t copulas. This may be because the Clayton-NSD copula does not exhibit extreme-value dependence, i.e. it fails the test of extreme-value dependence [55]. The skew-t copula might have performed not as well as the t copula due to over-parameterization. In addition, maximum likelihood estimation for the skew-t copula was extremely time consuming even for three dimensions and thus intractable for higher dimensions.

### B.5   Out-of-distribution detection

The inliers were generated from the Clayton-NSD copula representing the monthly rainfall of French Britanny and the outliers were generated uniformly at random on the unit cube. The number of inlier observations was $n_{in} = 225$, the number of outlier observations was $n_{out} = 25$, corresponding to 10% data contamination. A visual comparison of the results is given in Figure 11.

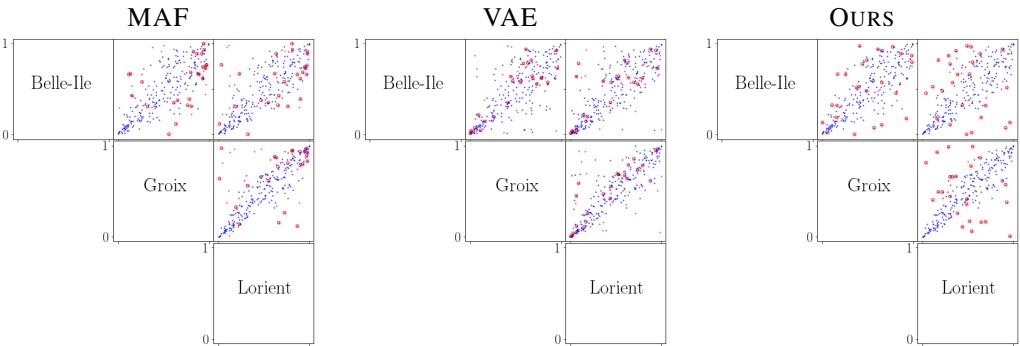

Figure 11: Out-of-distribution detection based on likelihoods. Inliers are represented with blue dots, outliers with red dots, and detected points are circled in red.

The masked autoregressive flow (MAF) provided an explicit likelihood. The likelihood of the variational autoencoder (VAE) was approximated from the reconstruction error. Including the KL divergence to the latent prior made results worse. The likelihood of the Archimax copula was approximated from the inclusion-exclusion scheme, checked to converge using various interval sizes.

### B.6   High dimensional modeling

We infer and sample a 100-dimensional Clayton-NSD Archimax copula, with parameters $\theta = 2$, $\alpha = (\alpha_0, \cdots, \alpha_0)$, $\alpha_0 = (1, 1, 1, 1, 2, 2, 2, 3, 3, 4)$, $\rho = 0.69$ using our method. The inference results are given in Table 2 of the main paper. We plot the samples in Figure 12, 20 coordinates at a time, with coordinates $(0, 1, 2, 3, 4, 5, 6, 7, 8, 9, 20, 21, 22, 23, 24, 25, 26, 27, 28, 29)$ in Figure 12 (a) and coordinates $(0, 10, 20, 30, 40, 50, 60, 70, 80, 90, 9, 19, 29, 39, 49, 59, 69, 79, 89, 99)$ in Figure 12 (b). The generated samples are in blue above the diagonal, the true samples are in black below the diagonal.

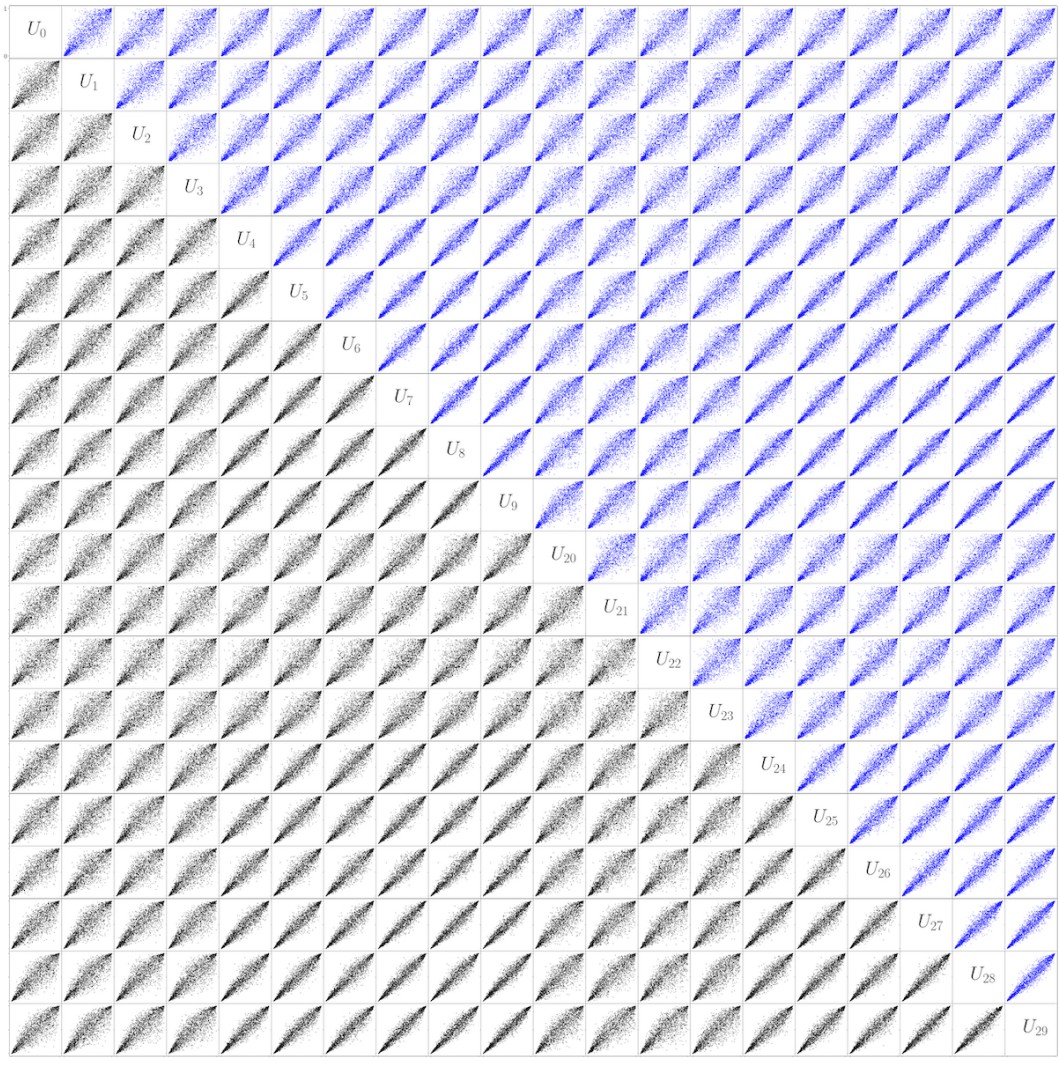

Figure 12: (a) Samples from 100-dimensional Clayton-NSD copula, showing coordinates $(0, 1, 2, 3, 4, 5, 6, 7, 8, 9, 20, 21, 22, 23, 24, 25, 26, 27, 28, 29)$.

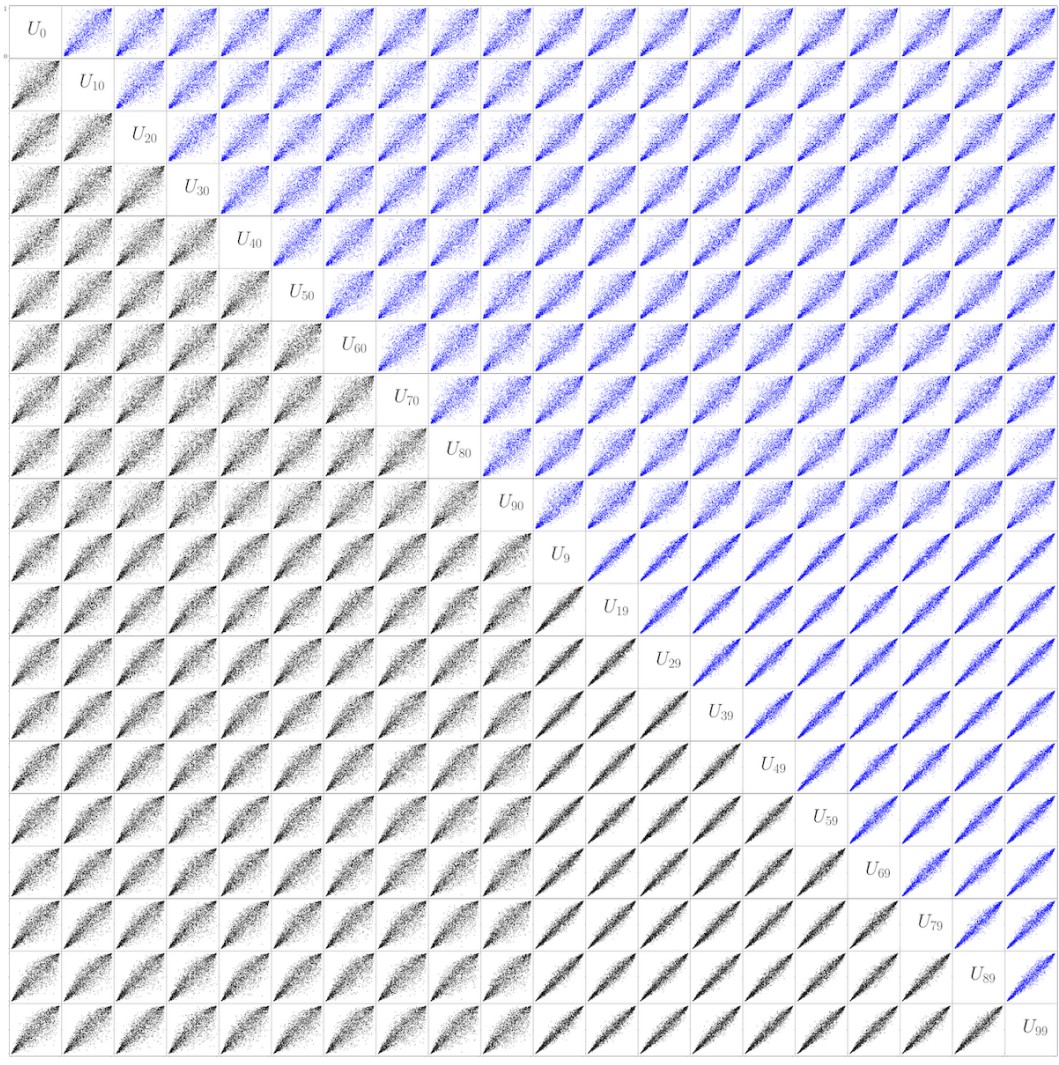

Figure 12: (b) Samples from 100-dimensional Clayton-NSD copula, showing coordinates $(0, 10, 20, 30, 40, 50, 60, 70, 80, 90, 9, 19, 29, 39, 49, 59, 69, 79, 89, 99)$.